



# Data-driven modeling of environmental factors influencing Arctic methanesulfonic acid aerosol concentrations

Jakob Boyd Pernov[1,a], William H. Aeberhard[2], Michele Volpi[2], Eliza Harris[2,b], Benjamin Hohermuth[3], Sakiko Ishino[4], Ragnhild B. Skeie[5], Stephan Henne[6], Ulas Im[7], Patricia K. Quinn[8], Lucia M. Upchurch[8,9], and Julia Schmale[1]

[1]Extreme Environments Research Laboratory, École Polytechnique Fédérale de Lausanne, Sion, Switzerland
[2]Swiss Data Science Center, ETH Zurich and École Polytechnique Fédérale de Lausanne, Switzerland
[3]Schroders Capital ILS, Zurich, Switzerland
[4]Institute of Nature and Environmental Technology, Kanazawa University, Kanazawa, Japan
[5]CICERO, Center for International Climate Research, Oslo, Norway
[6]Empa, Swiss Federal Laboratories for Materials Science and Technology, Dübendorf, Switzerland
[7]Department of Environmental Science/Interdisciplinary Centre for Climate Change, Aarhus University, Roskilde, Denmark
[8]Pacific Marine Environmental Laboratory, National Oceanic and Atmospheric Administration, Seattle, WA, USA
[9]Cooperative Institute for Climate, Ocean, and Ecosystem Studies, University of Washington, Seattle, WA, USA
[a]Now at: School of Earth and Atmospheric Sciences, Queensland University of Technology, Brisbane, Australia.
[b]Now at: Climate and Environmental Physics, University of Bern, Sidlerstrasse 5, 3012 Bern, Switzerland

*Correspondence to*: Jakob Boyd Pernov (Jakob.pernov@epfl.ch) and Julia Schmale (julia.schmale@epfl.ch)

**Abstract.**

Natural aerosol components such as particulate methanesulfonic acid ($MSA_p$) play an important role in the Arctic climate. However, numerical models struggle to reproduce $MSA_p$ concentrations and seasonality. Here we present an alternative data-driven methodology for modeling $MSA_p$ at four High Arctic stations (Alert, Gruvebadet, Pituffik/Thule, and Utqiaġvik/Barrow). In our approach, we create input features that consider the ambient conditions during atmospheric transport (e.g., temperature, radiation, cloud cover, etc.) for use in two data-driven models: a random forest (RF) regressor and an additive model (AM). The most important features were selected through automatic selection procedures and their relationships with $MSA_p$ model output was investigated. Although the overall performance of our data-driven models on test data is modest (max. $R^2 = 0.29$), the models can capture variability in the data well (max. Pearson correlation coefficient = 0.77), outperform the current numerical models and reanalysis products, and produce physically interpretable results.

The data-driven models selected features related to the sources, chemical processing, and removal of $MSA_p$ with specific differences between stations. The seasonal cycles and selected features suggest gas-phase oxidation is relatively more important during peak concentration months at Alert, Gruvebadet, and Pituffik/Thule while aqueous-phase oxidation is relatively more important at Utqiaġvik/Barrow. Alert and Pituffik/Thule appear to be more influenced by processes aloft than in the boundary layer. Our models usually selected chemical processing related features as the main factors influencing $MSA_p$ predictions, highlighting the importance of properly simulating oxidation related processes in numerical models.



## 1    Introduction

Natural marine biogenic aerosols, e.g., particulate methanesulfonic acid ($MSA_p$) are becoming an increasingly
important part of the Arctic climate system, especially during summer, due to sea ice retreat as well as changing environmental
conditions and circulation patterns (Willis et al., 2023), yet their environmental drivers remain understudied (Schmale et al.,
2021). Processes leading to natural aerosol emissions are affected by climate change, leading to ongoing changes in the natural
aerosol baseline. Understanding natural aerosols has implications for accurate modeling of the pre-industrial atmosphere and
thus estimation of the indirect aerosol effect (Carslaw et al., 2013; Menon et al., 2002). Natural aerosols, such as $MSA_p$, are
important seeds for low-level mixed-phase clouds in the Arctic (Abbatt et al., 2019; Beck et al., 2021). Low level clouds can
have a significant effect on the surface energy budget, influencing snow cover, sea-ice extent, and the Greenland ice sheet
behavior (Arouf et al., 2024; Wendisch et al., 2019). The current understanding of the Arctic climate system is limited,
including due to an insufficient representation of low-level Arctic mixed-phase clouds in large-scale models (Morrison et al.,
2012; Pithan et al., 2016; Taylor et al., 2022). The inadequate representation of aerosol particles acting as cloud condensation
nuclei and ice nucleating particles may partly explain the shortcomings of cloud representation in large-scale models
(Mauritsen et al., 2011; Stevens et al., 2018). While significant progress has been made (Abbatt et al., 2019; Shupe et al., 2022;
Wendisch et al., 2019, 2024), there are still important gaps in the current understanding and modeling efforts of natural Arctic
aerosols (Schmale et al., 2021).

In the Arctic atmosphere, $MSA_p$ mainly derives from the oxidation of natural, marine emissions of dimethyl sulfide
(DMS) (Barnes et al., 2006), although other sources can make minor contributions such as lakes, coastal tundra, melt ponds,
and biomass burning (Levasseur, 2013; Mungall et al., 2016; Park et al., 2019). $MSA_p$ has also been associated with biogenic
terrestrial sources in the mid-latitudes (Li et al., 2021; Zhou et al., 2021) as well as recently on Svalbard (Boreddy et al., 2024).
Arctic marine phytoplankton and algae produce dimethylsulfoniopropionate as an osmoprotectant (Yoch, 2002), which is
enzymatically cleaved to produce DMS (Andreae, 1990; Kettle et al., 1999). Seawater DMS emission is the main source of
marine biogenic sulfur in the atmosphere (Hulswar et al., 2022; Lana et al., 2011). Although the majority of DMS is oxidized
within seawater, a fraction is ventilated into the atmosphere where it is photochemically oxidized by OH, $O_3$, $NO_3$, and halogen
species via two pathways (addition or abstraction), both of which depend on temperature (Barnes et al., 2006; Jiang et al.,
2021; Shen et al., 2022). The atmospheric lifetime of DMS is on the order of 1-2 days (Breider et al., 2010; Lundén et al.,
2007), depending on latitude and environmental conditions (Ghahreman et al., 2019). DMS oxidation in the gas-phase proceeds
through several intermediates and ultimately yields MSA or $SO_2$. The addition pathway is more efficient at colder temperatures
(Shen et al., 2022) and results in a higher yield of gas-phase MSA (Sørensen et al., 1996). In the aqueous phase, dissolved
DMS (or its intermediates) is oxidized mainly by $O_3$ and OH, either through processes in cloud droplets or on deliquesced
particles, which is an important formation mechanism for $MSA_p$ (Baccarini et al., 2021; Chen et al., 2018; Fung et al., 2022;
von Glasow and Crutzen, 2004; Hoffmann et al., 2016; Kecorius et al., 2023; Wollesen de Jonge et al., 2021). $MSA_p$ can also
form via gas-phase oxidation of DMS and condensation of gaseous MSA. After formation in the aqueous phase, $MSA_p$ can be



released into the gas-phase during droplet evaporation and go on to further impact secondary aerosol production (Baccarini et al., 2021; Fung et al., 2022; Kecorius et al., 2023). Currently, the relative importance of gas- versus aqueous-phase oxidation of DMS is a topic of active research (Baccarini et al., 2021; Hoffmann et al., 2016; Wollesen de Jonge et al., 2021). The lifetime of $MSA_p$ is on the order of several days in the Arctic depending on the environmental conditions (Mungall et al.,

2018). MSA mainly resides in the accumulation mode (aerosols with a diameter > 100 nm) (Kerminen et al., 1997; Phinney et al., 2006; Xavier et al., 2022) although MSA can also be present in the Aitken mode (~25 < diameter < 100 nm) (Lawler et al., 2021) and makes a minor contribution to the coarse mode (> 1 µm) (Kerminen et al., 1997). Seasonally, $MSA_p$ displays near-zero values during the dark polar night with extensive sea ice coverage and little biological activity and the highest values during the sunlit, warmer polar day with retreating sea ice and highly biologically active waters (Sharma et al., 2012; Becagli

et al., 2016, 2019; Jang et al., 2021). Depending on location, maximum $MSA_p$ concentrations are reached during early, mid, or late summer, which are related to differences in atmospheric circulation patterns in relation to biologically active waters and marginal ice zones, microbiological differences in these sources regions that produce different DMS emissions, meteorological conditions (e.g., solar radiation and precipitation), and other environmental factors (different atmospheric oxidants and sea ice coverage) (Becagli et al., 2016, 2019; Moffett et al., 2020; Moschos et al., 2022; Nielsen et al., 2019;

Nøjgaard et al., 2022; Sharma et al., 2012, 2019). Dry and wet deposition are the main atmospheric removal mechanisms (with wet deposition making a larger contribution) as well as oxidation into sulfate (Chen et al., 2018; Fung et al., 2022).

The low accumulation mode particle concentrations characterize the summertime Arctic atmosphere as an aerosol-sensitive cloud condensation nuclei (CCN) regime (Birch et al., 2012; Mauritsen et al., 2011; Motos et al., 2023), thereby any variations in the number of CCN-active aerosols can have large consequences for the cloud radiative balance (Carslaw et al.,

2013). The low accumulation mode concentrations also create conditions conducive for new particle formation and growth. While modeling studies indicate MSA can participate in new particle formation (Chang et al., 2011; Li et al., 2024; Ning and Zhang, 2022), this has yet to be directly observed in the field (Beck et al., 2021; Dall'Osto et al., 2018), but has been demonstrated through chamber (Rosati et al., 2021) and flow tube studies (Johnson and Jen, 2023). Before these new particles can act as CCN they must first grow to sufficient sizes. MSA is especially critical for the condensational growth of aerosols to

CCN sizes (Ghahreman et al., 2019, 2021; Park et al., 2021) thereby affecting cloud microphysical properties such as cloud lifetime, albedo, and precipitation efficiency (Hansen et al., 1997; Ramanathan et al., 2001; Rosenfeld, 1999; Twomey et al., 1984). Elucidating the sources and atmospheric drivers of $MSA_p$ is crucial for reliable modeling of the Arctic climate system when considering that aerosol-cloud interactions are one of the largest sources of uncertainty in global climate modeling (Regayre et al., 2020).

The Arctic climate system is driven by many interconnected processes and feedback mechanisms making it difficult to disentangle the role of specific processes which is especially evident for aerosol-climate interactions (Schmale et al., 2021). Numerical modeling is currently the best method for exploring these complex processes and phenomena. Numerical models are defined here as global models, based on physical and chemical equations, used to simulate atmospheric composition and conditions. Numerical models can simulate Arctic aerosols, although some of the key underlying aerosol processes are often



simplified, approximated, or not represented due to lack of observations, unknown physical properties, or poorly parameterized mechanisms (Eckhardt et al., 2015; Emmons et al., 2015; Im et al., 2021; Monks et al., 2015; Whaley et al., 2022). There are also differences between models that create large uncertainties about future processes and their effects on aerosols, as well as aerosols' effect on Arctic climate. For instance, sea ice is drastically declining (Stroeve and Notz, 2018), and while models predict an increase in natural aerosols, they do not agree on the climate effects (Browse et al., 2014; Gilgen et al., 2018;

Struthers et al., 2011). Constraining numerical model uncertainty can be achieved by incorporating in situ observations (Regayre et al., 2020) but also through machine learning (or data-driven modeling, see below). This can be achieved through bias-correction methods (Lapere et al., 2023; Ran et al., 2023), using data-driven modeling algorithms to parameterize unresolved processes (Brajard et al., 2021; Yuval and O'Gorman, 2020), or combining data-driven modeling with ambient observations to model key atmospheric species and identify its drivers (Gilardoni et al., 2023; Hu et al., 2022). Improving the

skill of numerical models in the Arctic can greatly aid in our ability to understand, predict, and possibly mitigate the effects of climate change not only in the Arctic but globally, and data-driven modeling is an important avenue for accomplishing this.

       Data-driven models, coming from the statistical and machine learning literature, tend to rely less on prior knowledge of physical processes than numerical models and attempt to learn dependencies across data directly from some available observations. The rationale of "letting the data speak" is that a relevant relation across variables should in principle be found

with the appropriate amount of data and a proper representation of it, as long as the data-driven model is flexible enough and the signal-to-noise ratio is adequate (Breiman, 2001). As such, these data-driven models can confirm known processes and relations as well as potentially discover unknown ones. Such data-driven models can also be tailored to maximize out-of-sample prediction (e.g. forecasting in time) while retaining interpretability (Rudin et al., 2022). The general framework of non-linear regression appears appropriate for modeling and predicting complex environmental processes (Hastie et al., 2009) as

represented by heterogeneous data sources: the relation between the target variable (here $MSA_p$) and different input variables, hereafter referred to as features, can be approximated by training a data-driven model. Estimated relations can be ranked in terms of their contribution to the minimization of a loss function and non-relevant relations can be removed, making for more compact and parsimonious data-driven models and simplifying post-hoc interpretation. Any unexplained variability in the target variable, i.e., not captured by the approximated relations, is represented by an additive random error term. This class of

data-driven models includes (generalized) additive models (Hastie and Tibshirani, 1990) as well as variants and extensions of regression trees (Breiman et al., 1984), among others. Additive models (AM), and generalized additive models (GAMs) more broadly, are fairly established for empirical modeling in various fields such as ecology, epidemiology, and Earth sciences when the interpretability of results is important (Wood, 2017; Zuur et al., 2009). In climate science and meteorology, GAMs are often used for spatial interpolation (Aalto et al., 2013; Pearce et al., 2011) and simulating sources of atmospheric constituents

(Yue et al., 2023). Machine learning models like a random forest (RF) are increasingly recognized to outperform AMs/GAMs in terms of out-of-sample prediction (Bonsoms and Ninyerola, 2024). Nonetheless, some recent studies still advocate for the benefit of easily identifying drivers of natural phenomena, and directly interpreting their effect, with AMs/GAMs (Deger et al., 2024; Gao et al., 2023), highlighting their applicability to this study. RF models have been utilized for investigations of



environmental phenomena. Song et al. (2022) used a random forest regressor to investigate the drivers of different aerosol types on Svalbard with accurate results ($R^2 = 0.79$) and found that solar radiation, surface pressure, and temperature were drivers of biogenic-type aerosols (which contained high amounts of MSA). Nair and Yu (2020) trained an RF model on long-term simulations of a global size-resolved particle microphysics model (GEOS-Chem-Advanced Particle Microphysics) to simulate cloud condensation nuclei concentrations, which was robust and accurate. Overall, these studies highlight the applicability of RF regressor and additive models in understanding complex atmospheric phenomena.

Modeling natural aerosol processes in the Arctic remains a challenge but is critical to investigating the energy balance of this fast-changing, pristine region. In this study, we aim to (1) evaluate the performance of numerical models at simulating $MSA_p$ in the Arctic, (2) develop a data-driven methodology to simulate the seasonal cycle of $MSA_p$ at various locations, and (3) investigate the environmental drivers of $MSA_p$. The study is structured in the following manner:

- In Sect. 2, we describe the input data (Sect. 2.1, in situ observations, reanalysis products, satellite, and numerical model output), feature engineering procedure (Sect. 2.2), preparation of input data (Sect 2.3, temporal aggregation, feature grouping, and multi-site merging), model performance evaluation (Sect. 2.4), and data-driven models (model details, feature selection procedure, and model interpretation).

- In Sect. 3, we analyze the seasonal cycles of in situ $MSA_p$ at the High Arctic stations (Sect. 3.1), evaluate the current performance of numerical models (Sect. 3.2), and our data-driven models at simulating $MSA_p$ at each station (Sect. 3.3), and lastly explore the features selected by the models as being important for MSA production (Sect. 3.4) and how they affect model output of $MSA_p$ (Sect. 3.5).

We show that existing numerical models struggle to reproduce the seasonal cycles and magnitudes of $MSA_p$ compared to observations, however, investigation of the underlying causes of these discrepancies is beyond the scope of this work. Our data-driven models outperform the numerical models although the evaluation metrics are modest at best. The data-driven models select features related to the source and chemical processing of MSA precursors as well as $MSA_p$ removal, indicating the data-driven models give physically interpretable results. While both gas- and aqueous-phase oxidation are likely occurring at all sites, the seasonal cycles and selected features suggest that during *peak concentration months* gas-phase oxidation is more relatively important at Alert, Gruvebadet, and Pituffik/Thule while aqueous-phase oxidation is more relatively important at Utqiaġvik/Barrow. Results also indicate that Gruvebadet and Utqiaġvik/Barrow are more influenced by surface-related processes compared to Alert and Pituffik/Thule which are more influenced by processes aloft.



## 2   Methods

### 2.1   Datasets

#### 2.1.1     In situ aerosol observations

In situ filter samples of particulate methanesulfonic acid ($MSA_p$) were measured at four Arctic stations (Alert, Gruvebadet, Pituffik/Thule, and Utqiaġvik/Barrow) (Becagli et al., 2016, 2019; Moffett et al., 2020; Sharma et al., 2019). Figure 2a displays the location of each station and details about each station are given in Table 1. For Alert, Gruvebadet, and Pituffik/Thule, samples from 2010-2017 were used as each site contained sufficient data coverage and a consistent sampling frequency, while for Utqiaġvik/Barrow, samples included 2008-2014 due to data availability and changes in sampling frequency (Moffett et al., 2020). Details about the analytical instrumentation and methods are described in Supplementary Text 1. While there are differences in sampling (different inlet and temporal resolution) and analysis (different ion chromatographs) at each station, these measurements are considered comparable as an analysis by two different laboratories for samples from Alert in 2018 showed good agreement (Moschos et al., 2022) and ion chromatography is a reproducible methodology (Xu et al., 2020).

**Table 1: Details of the four Arctic stations.**

| Station Name | Latitude | Longitude | Altitude (m asl) | Sampling Frequency (days) |
|---|---|---|---|---|
| Alert | 82.5° N | 62.4° W | 210 | 7 |
| Gruvebadet | 78.9° N | 11.9° E | 50 | 1 |
| Pituffik/Thule | 76.5° N | 68.8° W | 220 | 2 |
| Utqiaġvik/Barrow | 71.3° N | 156.6° W | 10 | 1-5 |

#### 2.1.2     ERA5

ERA5 is the fifth-generation atmospheric reanalysis product from ECMWF (Hersbach et al., 2020), based on the Integrated Forecast System (IFS) cycle 41r2 numerical model. In this study, ERA5 data on a 0.5° × 0.5° resolution for north of 45 °N and every third hour was used to match the geographical extent and temporal resolution of the output derived from the atmospheric transport model FLEXPART (Sect. 2.1.3). Surface-level (SL) and vertically resolved ERA5 data on model levels (ML) were used. The height of each model level on each grid cell was converted to geopotential height using the vertically resolved temperature and specific humidity as well as the logarithm of the surface pressure and the surface geopotential. Relative humidity was calculated using 2m air temperature and dew point temperature following the method of Pernov et al. (2024a). Here we use ERA5 data from April 1 to September 30 for 2008-2017. Recently, ERA5 surface level variables were compared against continental ground-based stations spanning at least 1 decade for most sites. Overall ERA5 performed well for temperature, solar radiation, and pressure although less so for relative humidity and wind speed/direction (Pernov et al., 2024a). ERA5 is one of the best reanalysis datasets for reproducing precipitation (Loeb et al., 2022) and has



shown skill in reproducing precipitation for various regions (Bandhauer et al., 2022; Beck et al., 2019) as well as for the Arctic (Handong et al., 2021). Overall, these limitations should not affect the use of ERA5 or our interpretations. The ERA5 variables were selected based on domain knowledge of the atmospheric conditions which could plausibly affect DMS emission,

oxidation to MSA, and removal of MSA aerosols. These include oceanic variables such as sea ice concentration (used to filter ocean biology features, see below) and sea surface temperature, physical atmospheric variables such as wind speed (WS), temperature at the surface (T2M), boundary layer (T_BL), and free troposphere (T_FT), shortwave and longwave downwelling radiation (SSRD and STRD, respectively), boundary layer height (BLH), and hydrological atmospheric variables such as relative humidity (RH), specific humidity (Q), low cloud cover (LCC), large-scale rain rate (LSRR), total column cloud liquid

water content (TCLW), and specific cloud liquid water content (LWC). Table 2 lists more details about the ERA5 variables used in this study.

### 2.1.3  FLEXPART

Air mass residence times were simulated with the Lagrangian particle dispersion model FLEXPART v9.1 (Pisso et al., 2019), driven with meteorological data from the ERA5 reanalysis with 0.5° x 0.5° resolution and 137 vertical levels available every

three hours. ERA5 data for FLEXPART were obtained using the Flex extract package (Tipka et al., 2020). 50,000 passive air tracer model particles, representing a passive air tracer without removal processes, were released every three hours at each of the atmospheric observatories and tracked for up to 10 days backward in time with an output frequency of three hours. The vertical limit of the FLEXPART output was 15,000 m. For Alert, Pituffik/Thule, and Utqiaġvik/Barrow, a release height of 10 m above ground level (agl) was used. For Gruvebadet, to account for the complex topography, a range of 10-100 m agl was

used as the release height. The main output from FLEXPART consists of 3-dimensional fields of residence time in units of seconds (s). In contrast to Eulerian models, Lagrangian dispersion models can be applied in time-reversed mode and superior in representing plumes emerging from point releases (Pisso et al., 2019). However, the quality of their results can be limited by the offline nature of the coupling to meteorological fields, which are restricted in spatial and especially temporal resolution (Brioude et al., 2013). The FLEXPART output was combined (Sect. 2.2) with other data sources for calculating additional

input variables for the data-driven models. FLEXPART residence time was combined with boundary layer height from ERA5 to calculate the residence time air masses within the boundary layer (RT_BL) or free troposphere (RT_FT). Sea ice concentrations from ERA5 were combined with FLEXPART to calculate the residence time of air masses over open water (OPEN_WATER, sea ice concentration < 20 %), open pack ice (OPEN_PACK_ICE, > 20 % and < 80 %), and consolidated pack ice (CONSOLIDATED_PACK_ICE, > 80 %), which was normalized by the grid cell area to give units of s km$^{-2}$. The

precipitation type from ERA5 (no precipitation, rain, freezing rain, snow, wet snow, mixture of rain and snow, ice pellets) was combined with FLEXPART to calculate the residence time of air masses experiencing no precipitation (NO_PRECIP) or precipitation (sum of the amount of time air masses experienced any precipitation types, PRECIP) which was normalized by the grid cell area to give units of s km$^{-2}$.



### 2.1.4    CAMS

The Copernicus Atmosphere Monitoring Service Re-Analysis dataset (hereafter referred to as CAMS) is the latest reanalysis product produced by ECMWF, including three-dimensional fields of meteorological variables, chemical, and aerosol species for the period from 2003 onwards. CAMS data was obtained from the Copernicus Atmospheric Data Store (ADS) (https://ads.atmosphere.copernicus.eu/cdsapp#!/home, last accessed 08/11/2022). CAMS is based on the ECMWF's IFS CY42R1 cycle and the 4D-VAR data assimilation system (Inness et al., 2019) and uses an extended version of the Carbon

Bond 2005 (CB05) tropospheric chemical mechanism (Flemming et al., 2015). Emissions consist of MACCity (MACC and CityZEN EU projects) anthropogenic emissions (Granier et al., 2011), GFAS (Global Fire Assimilation System) fire emissions (Kaiser et al., 2012), and MEGAN2.1 (Model of Emissions of Gases and Aerosols from Nature) biogenic emissions (Guenther et al., 2006). The CAMS data have a spatial resolution of 0.75° × 0.75° with 60 hybrid sigma–pressure (model) levels (13 levels between approximately 400 and 100 hPa) in the vertical (top level at 0.1 hPa) and a temporal resolution of 3 h. The two

oxidants, ozone ($O_3$) and the hydroxyl radical (OH) in the boundary layer and free troposphere were used from CAMS as they are related to the gas- and aqueous-phase oxidation of DMS and its intermediates to MSA (Barnes et al., 2006). CAMS output of $MSA_p$ was extracted using the nearest grid cell to the stations' location (Table 1) for the lowest level and converted from mass mixing ratio to mass concentration using the ambient temperature and pressure from CAMS for comparison to numerical models. CAMS output of $MSA_p$ was not included in the data-driven models. To match the spatial resolution of different

datasets, re-gridding, using bilinear interpolation from the xESMF (v0.8.2) python package (Zhuang et al., 2023) was applied to the FLEXPART dataset to match the CAMS spatial resolution.

### 2.1.5    Chlorophyll-a

Chlorophyll-a (ChlA) is commonly used as a proxy for phytoplankton biomass and oceanic productivity (Arnold et al., 2010; Huot et al., 2007), and was included for that purpose in this study. Level 3 datasets of satellite-derived daily surface

chlorophyll-a concentration with a spatial resolution of 4 km from the European Space Agency's GlobColour Project3 (https://www.globcolour.info/, last access October 1, 2022) were obtained from the Copernicus Marine Environment Monitoring Service (CMEMS4). This product is produced by reprocessing the merged observations from five satellite radiometers (OLCI from Sentinel 3a and 3b, MODIS on Aqua, and VIIRS from Suomi-NPP and JPSS-1), therefore missing data due to the presence of clouds is minimized. The GlobColour dataset is a common and suitable choice for investigating

phytoplankton (Ardyna et al., 2017; Becagli et al., 2022; Cole et al., 2015; Xi et al., 2020). The ChlA datasets were re-gridded using bilinear interpolation (xESMF v0.8.2 python package (Zhuang et al., 2023)) to match the 0.5° spatial resolution of FLEXPART.



### 2.1.6    DMS Flux

Oceanic emissions of dimethyl sulfide (DMS) were used to evaluate the ocean-air exchange of DMS and were downloaded
from the Copernicus ADS webpage ([https://ads.atmosphere.copernicus.eu/cdsapp#!/dataset/cams-global-emission-inventories?tab=overview](https://ads.atmosphere.copernicus.eu/cdsapp#!/dataset/cams-global-emission-inventories?tab=overview), last access 15 September 2022) and were not calculated offline for this study. DMS is the initial
precursor for MSA formation, therefore, information on its oceanic emission is central to investigating processes related to
MSA variation. The estimation of oceanic DMS emissions to the atmosphere requires DMS concentrations in the ocean as
well as meteorological variables, specifically the u and v components of 10-meter wind speed, as well as the sea surface
temperature. The oceanic DMS concentrations used for the flux estimation were provided by Lana et al., (2011). The data are
derived from numerous measurements obtained for the period 1989-2009 and were obtained from the Surface Ocean Lower
Atmosphere Study (SOLAS) webpage ([https://www.bodc.ac.uk/solas_integration/implementation_products/group1/dms/](https://www.bodc.ac.uk/solas_integration/implementation_products/group1/dms/), last
access 15 September 2022). It should be noted that these oceanic DMS concentrations are based on a monthly climatology.
Formulas for the calculation of the DMS flux were provided by Nightingale et al. (2000). Meteorological data computed by
the Norwegian Meteorological Institute using the ECMWF-IFS model version Cy40r1 were used. The daily mean emission
data are provided on a regular longitude-latitude grid on $0.5° \times 0.5°$ resolution for the period 2000 to 2018.

### 2.2   Feature Engineering: Residence time weighted average of environmental variables

For our data-driven modeling efforts, we engineered appropriate input features to capture the air mass history (environmental
conditions and surface interactions) in a time-resolved manner, i.e., capturing the environmental conditions where an air mass
was actually located for different intervals backward in time. To create a time-resolved air mass history, FLEXPART residence
time and environmental variables from the datasets described in Section 2.1 were combined. A total of five timesteps backward
in time was selected as the duration of the air mass history: as the lifetime of DMS in the atmosphere is approximately 2 days
(Breider et al., 2010; Lundén et al., 2007), this can account for the emission and oxidation of DMS and the detection of MSA
at the ground-based stations. Daily intervals were selected as the temporal resolution of this air mass history as a compromise
between high enough time resolution to capture physical and chemical processes and the number of input features in our
models. We also selected daily resolution for the time-resolved air mass history to match the highest sampling frequency (daily
at Gruvebadet). For each variable and observation, we calculated aggregations for daily intervals (up to 5 daily timesteps
before release time) backward in time as indicated in Table 2. For the vertically resolved environmental variables (ERA5 and
CAMS), the geopotential height of each grid cell was calculated according to the ERA5 documentation using temperature,
surface level pressure, and geopotential height (IFS Documentation CY41R2 - Part III: Dynamics and Numerical Procedures,
2024). This geopotential height of each grid cell was compared to the boundary layer height from ERA5. Grid cells inside the
boundary layer were averaged to create a boundary layer average of the environmental variables. Grid cells above the boundary
layer height were averaged up to the ERA5 model level corresponding to the highest non-zero FLEXPART level to create a
free troposphere average of the environmental variables. The residence time in the boundary layer and free troposphere was



calculated by summing the FLEXPART residence time over all longitudes and latitudes for grid cells below or above the boundary layer height, respectively. The relative residence time (boundary layer or free troposphere) was calculated by normalizing the FLEXPART residence time in each grid cell to the sum of FLEXPART residence times over all grid cells and was applied to the boundary layer and free troposphere separately. To account for different sized grid cells, the relative FLEXPART residence time was weighted by the area of each grid cell (grid cell area weighted relative residence time). The

grid cell area weighted relative residence time was used to calculate a weighted average of the environmental variables. In this manner, we could ascertain the environmental conditions while accounting for where air masses actually were, directly accounting for transport at our locations of interest. A schematic for the feature engineering procedure is displayed in Fig. 1 using SSRD at Gruvebadet on 1 June 2010 as an example.

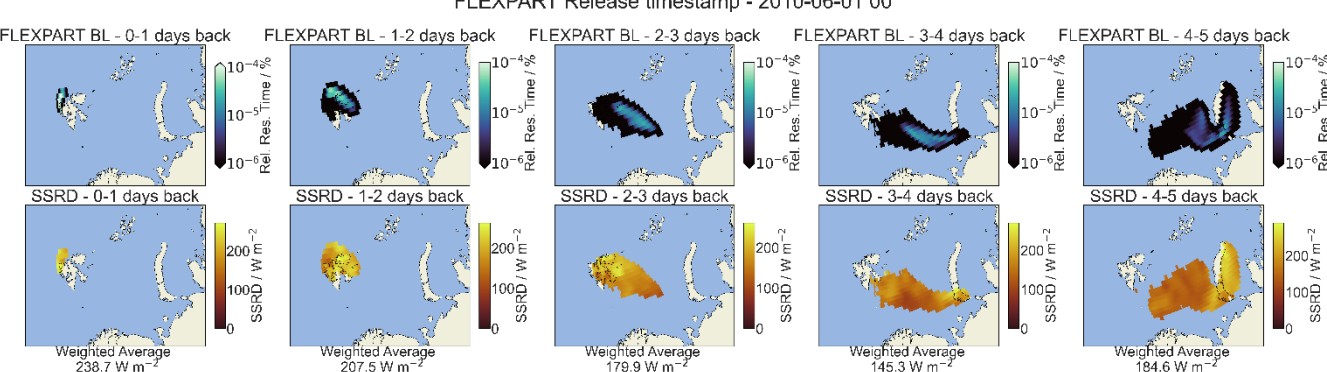


**Figure 1. Schematic of the feature engineering process.** The top row represents the relative FLEXPART boundary layer residence time and the bottom row shows the average surface solar radiation downwards (SSRD, Table 1) for the different daily intervals backward from 2010-06-01 00 for Gruvebadet. Calculating a weighted average of the SSRD using the relative residence time as weights results in the weighted average listed below each SSRD subpanel.

**2.3  Preparation of input data**

Measurements of $MSA_p$ at the ground-based stations varied in terms of frequency and regularity, while the feature-engineered variables (described above in Sect. 2.2) were initially processed at hourly resolution for every third hour (the temporal resolution of the FLEXPART output). The variables therefore needed to be temporally aggregated to match the station measurements. The aggregation was done over non-overlapping time windows corresponding to the sampling periods of each installed aerosol filter. For this aggregation, some features were summed while others were averaged, according to the

physical nature of each variable and how it relates to MSA formation/removal (see Table 2 for more details). For instance, time over open water (OPEN_WATER) was summed as the total amount of time air masses spent over open water is more informative than an average, whilst for the 2-meter temperature (T2M), a sum is not physically meaningful therefore an arithmetic mean was applied. LSRR (originally expressed as mm day$^{-1}$ in ERA5) was summed over the daily intervals to give





units of mm. Total DMS emission is originally expressed as kg m$^{-2}$ s$^{-1}$, during the feature engineering procedure the time unit was converted to days, the area unit was converted to km$^{-2}$, the emission was summed over the daily intervals, normalized to the grid cell area, and summed over all grid cells for a given daily interval to give units of kg (which was then summed over the filter collection period).

The four stations only measure MSA$_p$ concentrations locally, therefore, models were first trained and tested on the
specific stations individually, as indicated by "St" throughout the text. To model Pan-Arctic MSA$_p$, we created two additional datasets to train our models. The first one is called All Stations Full (ASF), which is simply the merger of all data from the four stations. For this, the stations' geographical coordinates were not used: Stations were implicitly considered as independent replicates (in a statistical sense) if they had data on the same day. The second additional dataset is called All Stations (AS), which is another merger of a subset of data from the four stations: We sub-sampled measurements from the stations with higher
temporal frequency (e.g., Gruvebadet with mostly daily measurements) to match that of the lowest temporal frequency (Alert, with roughly weekly measurements). Therefore, in AS all four stations are represented equally in terms of the number of observations.

The feature engineering presented in the previous sections produced a large number of variables we could include in our models as predictors. The different data sources also had varying degrees of accuracy and reliability. We therefore
manually subset the features into two groups, denoted as Group A and B. Group A included the variables that we deemed to be the most related and reliable among the predictors of MSA$_p$, using domain knowledge of atmospheric chemistry and physics. For instance, surface air temperature affects the oxidation pathways of DMS and the thermodynamic phase of water in the atmosphere furthermore this variable is well reproduced by ERA5 in the Arctic (Pernov et al., 2024a), hence was included in Group A. Group B includes features which were expected to be good predictors for MSA, although the accuracy of these
variables may be lower in the areas covered by our study. For instance, measurements of hydroxyl radical mixing ratios (OH) are analytically challenging and datasets are sparse (Lelieveld et al., 2016; Stone et al., 2012), therefore CAMS cannot be validated against sufficient in situ observations, especially in the Arctic, hence it was included in Group B. DMS flux is based on a monthly climatology of seawater DMS concentrations (Lana et al., 2011), therefore, short-term variations depend only on parameterizations based on wind speed and sea surface temperature, and hence was included in Group B. Table 2 lists all
features in Groups A and B. Table A1 lists commonly used abbreviations throughout this manuscript.

## 2.4  Model evaluation

We evaluated our models by assessing the out-of-sample prediction error. To this end, we first performed a training-test split: for every station, we left out some observations corresponding to one or two summers, before attempting any modeling (Table 3). These were our test subsets, and they were used to assess prediction error as a last step for the final versions
of the models presented below. The remaining data are our training subsets on which we applied a temporal cross-validation (CV) scheme. This CV scheme was mainly used for hyperparameter tuning for the baseline models (see Section 2.6.1) and was the criterion in the feature selection procedure for the additive model (see Section 2.6.2). We used a six-fold CV,



corresponding to leaving out one year of data from the training set (between 2010 and 2015, see Table 3) for each station. Thus, five years of data were used for fitting the models and out-of-sample prediction could be performed on the one year of

data in the left-out fold. Details about both training and test data for all stations are summarized in Table 3. Among other accuracy metrics, CV-based mean squared error (MSE) was computed as an average over the six folds. MSE is defined by Eq. (1):

$$MSE = \frac{\sum_{i=1}^{n}(y_i - \hat{y}_i)^2}{n} \tag{1}$$


where $y_i$ is an observation and $\hat{y}_i$ is the prediction of the model on this datapoint (from either RF or AM), and *n* stands for the number of observations in a given fold for a given station. MSE values lie within $[0, \infty]$, where a value closer to 0 represents better predictions (lower error). Another two important metrics we report are the prediction coefficient of determination (or $R^2$ value) as defined by Eq. (2):


$$R^2 = 1 - \frac{\sum_{i=1}^{n}(y_i - \hat{y}_i)^2}{\sum_{i=1}^{n}(y_i - \bar{y}_i)^2} \tag{2}$$

where $\bar{y}$ denotes the mean of the observations in all other folds for a given station (constant prediction). We also report the Pearson (linear) correlation coefficient (PCC) as defined by Eq. (3):


$$PCC = \frac{\sum_{i=1}^{n}(y_i - \bar{y})(\hat{y}_i - \bar{\hat{y}})}{\sqrt{\sum_{i=1}^{n}(y_i - \bar{y})^2 \sum_{i=1}^{n}(\hat{y}_i - \bar{\hat{y}})^2}} \tag{3}$$

where $\bar{\hat{y}}$ denotes the mean of the predictions. Note that the $R^2$ can take values within $(-\infty, +1]$: A value of 0 means that the model prediction is equivalent to the average of the MSA values in the training set, a negative value means that the model

predictions are worse than this average, and a value closer to 1 means that the model predicts better than the training set average (a value of 1 meaning perfect prediction). It should be noted that the $R^2$ metric we use in this study is not the square of the PCC. The PCC is calculated using the stats module from the Python package scipy.

We compute all metrics on two scales: the original scale of values and the natural logarithm scale, the one used to train the models. The purpose of training the models and assessing their prediction on the log scale as well is that large

observations are compressed by the transformation, thus squared errors on the log scale may be more informative for the majority of the observations (i.e., less sensitive to potential outliers). The same metrics were also computed on the test set.





**Table 2: Key details of the features used for data-driven modeling of MSA$_p$.** Variables for the boundary layer and free troposphere are denoted by "BL" and "FT", respectively. ERA5 data on surface and model levels are denoted by "SL" and "ML", respectively. For the Aggregation Method column, "Average" indicates the arithmetic mean.

| Abbreviation | Description | Units | Dataset | Aggregation method | Group |
|---|---|---|---|---|---|
| WS_BL | Wind speed BL | m s$^{-1}$ | ERA5 ML | Average | A |
| WS_FT | Wind speed FT | m s$^{-1}$ | ERA5 ML | Average | A |
| OPEN_WATER | Time over open water (<20 % sea ice) | s km$^{-2}$ | ERA5 SL | Sum | A |
| OPEN_PACK_ICE | Time over open pack ice (>20 and <80 % sea ice) | s km$^{-2}$ | ERA5 SL | Sum | A |
| CONSOLIDATED_PACK_ICE | Time over consolidated pack ice (<80 % sea ice) | s km$^{-2}$ | ERA5 SL | Sum | A |
| RT_BL | Residence time BL | s | FLEXPART and ERA5 | Sum | A |
| RT_FT | Residence time FT | s | FLEXPART and ERA5 | Sum | A |
| SP | Surface pressure | hPa | ERA5 SL | Average | A |
| SST | Sea surface temperature | K | ERA5 SL | Average | A |
| Q_BL | Specific humidity BL | kg kg$^{-1}$ | ERA5 ML | Average | A |
| Q_FT | Specific humidity FT | kg kg$^{-1}$ | ERA5 ML | Average | A |
| T_BL | Temperature BL | K | ERA5 ML | Average | A |
| T_FT | Temperature FT | K | ERA5 ML | Average | A |
| T2M | Air temperature at 2 m | K | ERA5 SL | Average | A |
| SSRD | Solar shortwave radiation downwards | W m$^{-2}$ | ERA5 SL | Sum | A |
| STRD | Solar thermal radiation downwards | W m$^{-2}$ | ERA5 SL | Sum | A |
| ChlA | Chlorophyll A | mg m$^{-3}$ | Chlorophyll-a | Average | B |
| DMS | DMS emitted | kg | DMS Flux | Sum | B |
| TCLW | Total column cloud liquid water | kg m$^{-2}$ | ERA5 SL | Average | B |
| O$_3$_BL | Ozone mixing ratio BL | ppbv | CAMS | Average | B |
| O$_3$_FT | Ozone mixing ratio FT | ppbv | CAMS | Average | B |
| LWC_BL | Specific Cloud Liquid Water BL | kg kg$^{-1}$ | ERA5 ML | Average | B |
| LWC_FT | Specific Cloud Liquid Water FT | kg kg$^{-1}$ | ERA5 ML | Average | B |
| BLH | Boundary Layer height | m | ERA5 SL | Average | B |
| OH_BL | OH radical mixing ratio BL | ppbv | CAMS | Average | B |
| OH_FT | OH radical mixing ratio FT | ppbv | CAMS | Average | B |
| LCC | Low cloud cover | (0-1) | ERA5 SL | Average | B |
| RH | Relative humidity | % | ERA5 SL | Average | B |
| PRECIP | Time with precipitation | s km$^{-2}$ | ERA5 SL | Sum | B |
| NO_PRECIP | Time with no precipitation | s km$^{-2}$ | ERA5 SL | Sum | B |
| LSRR | Large-scale rain rate | mm | ERA5 SL | Sum | B |

### 2.5 Imputing missing values

Missing data for both the in situ MSA$_p$ measurements (target variable) as well as for the input variables (features) exist and potentially could affect or bias our analyses. Regarding the in situ MSA$_p$ measurements, we considered the station-specific aerosol filter collection duration (called hereafter nominal resolution) as a reference over which features were aggregated. These nominal resolutions were: daily for Gruvebadet and Utqiaġvik/Barrow, every two days for Pituffik/Thule, and seven days for Alert. Based on a trial and error approach, we decided to enforce the rule that any sequence of consecutive missing values longer than three times this nominal resolution would be deemed too long to be imputed without introducing artifacts. These long patches were thus left as is and features were aggregated over time windows according to the nominal resolution. Shorter sequences of consecutive missing values were imputed at the nominal resolution. For Gruvebadet and Pituffik/Thule,



this was done by linear interpolation using the two closest available measurements. For Utqiaġvik/Barrow, the variable temporal resolution depending on the time of year (Table 1) complicated this procedure, and gaps of three and four days occurred too often for our rule to be applied strictly at a daily nominal resolution. Here we left gaps up to five days (as these could be valid measurements) as is and imputed by linear interpolation based on the two closest values to those gaps lasting

between five and ten days. Finally, Alert required more care, as missing values could last for long periods (> 3 weeks), making linear interpolation unreliable. Here, we used different imputation methods for short gaps (up to two missing values) and long gaps (three weekly values missing), targeting at most ten days between values. For short gaps, we used local quadratic fits, fitted by minimizing the sum of squared residuals on the natural logarithm scale. We used neighborhoods of three available values before and after the gaps, weighted by a Gaussian kernel. For the single long gap, we used a model with a polynomial

of degree 5 representing long-term time trends and yearly seasonality represented by a linear combination of cubic $B$-splines, also fitted by minimizing the sum of squared residuals on the log scale. Figure S14 illustrates the imputation of such short and long gaps for Alert in situ measurements.

Regarding the input feature, ChlA, to minimize the impact of short gaps due to clouds or the presence of sea ice, we studied different data imputation strategies. We first assessed seven different algorithms (mean, median, imputeTS (Moritz

and Bartz-Beielstein, 2017), $k$ nearest neighbor, principle component analysis, and MissForest) based on randomized masking of measurements for Alert and measured the reconstruction error over the imputed values. Within the feature set, there are strong correlations that can be exploited to fill measurements. We found that MissForest (Stekhoven and Bühlmann, 2012) was the best-performing method, and we used this to impute values for the entirety of the feature input dataset. MissForest is based on the application of Random Forests iteratively. First, it imputes missing input data using the mean. Then it trains a

Random Forest regressor on a set of fixed features, to predict missing values on a separate feature to be filled. It proceeds iteratively and stops when the predicted missing values converge, or when the maximum number of iterations is reached. MissForest is highly flexible and does not make any assumptions about the data distribution. However, purely statistically driven data imputation might lead to physically implausible values. To achieve consistency, we set all ocean biology variables (DMS and ChlA) to 0 if the sea ice concentration from ERA5 was >80 % as no ocean-atmosphere exchange is expected for

these conditions. For each station, measurements below the reported limit of detection were imputed with half the detection limit (Becagli et al., 2016, 2019; Moffett et al., 2020; Sharma et al., 2019).

**Table 3: Train/test splits for all stations.** N is the number of observations in each set.

|  | Training set years | $N$ | Test set years | $N$ |
|---|---|---|---|---|
| Alert | 2010-2015 | 166 | 2016-2017 | 56 |
| Gruvebadet | 2010-2015 | 937 | 2017 | 173 |
| Pituffik/Thule | 2010-2015 | 360 | 2016-2017 | 107 |
| Utqiaġvik/Barrow | 2010-2015 | 311 | 2008-2009 | 109 |



### 2.6 Data-driven models

For this task, we considered non-linear regression models approximating the log-transformed target, $MSA_p$ concentration, plus a constant as $Y_i = \ln(MSA_i + 10^{-3})$, for $i = 1,2,\ldots,N$, where $N$ is the sample size (different for each station, Table 3). Our choice of log-transformation and addition of a constant was based on achieving a somewhat symmetrical target distribution, which is better suited when using a mean squared error loss function, as well as improving numerical stability in the optimization. All models make use of the same engineered features presented above as inputs to predict $Y$. We considered two main approaches for modeling these relationships: a "baseline model" composed of a common random forest (RF) regressor (Breiman et al., 1984), which is a standard and well-accepted regression model, also offering some insights on feature importance; and we developed a specific additive model (AM), which models in a more principled manner the temporal relationships across the features and the target while providing a more interpretable model overall. The interpretability of estimated effects in the AM is a key aspect here, and the main reason why we developed it. The goal is to identify drivers and describe their relation with the target variable, while at the same time to have full control over the optimization process and variable selection procedure. We present the baseline RF model and its setup in the following Section 2.6.1 and the AM in Section 2.6.2. Other modeling approaches were explored, we summarize their performance in Supplementary Text 2 and Fig. S1. These other approaches were not retained because their predictive performance was no better than that of RF and AM. In the case of similar performance, RF and AM still had interpretability benefits, notably in identifying which features contributed the most to the model prediction power, and thus were the ones we retained.

#### 2.6.1    Baseline model: Random Forest

Random Forests (RF) are among the top-performing models in a wide variety of classification and regression tasks and are known to be robust to overfitting while being fast to train and fast at inference (Biau and Scornet, 2016). RFs are often a nominal selection for most data-driven applications. RFs are composed of an ensemble of decision trees, where each tree is trained on a random subset of data (a bootstrap) and by testing a random subset of features for each decision tree node optimizing an impurity measure. Averaging the output of each trained tree, allows the RF to predict a given input datapoint. In addition, RFs provide an implicit ranking of features, which for regression tasks is based on the average reduction in the squared error at node splits for a given feature, which we will refer to as an importance score. Although ranking features according to their explicit relationship with the target variable is a difficult problem, RFs provide a simple yet effective way to sort features from more to less important. This will be used to qualitatively compare with the selected features based on our proposed AM described in Section 2.6.2.

For each experiment with RFs, we performed a grid search for the depth of each tree and for the minimum number of data points per node to make it a leaf. Those two hyperparameters control how much each tree in the random forest can grow, trading off training accuracy for speed as well as avoiding overfitting. The number of trees was set to 500 and kept constant for all the experiments; a larger number of trees did not result in better models but only in increased computational time.





We selected the most important features for the RFs using a method analogous to the additive model forward selection procedure described in Section 2.6.2. First, for each of the 500 trees in each RF model, the list of features with a model importance score ≥ 5% of the maximum importance score for that tree was found. We then took the summed importance scores

for each feature across all trees in which they were selected and divided them by the total number of trees (500) to estimate the mean score of each feature only from the trees where it was selected. If this mean score was ≥ 5% of the mean of the maximum importance scores for each tree, the feature was selected for that model. Re-training the RF with only the selected features did not materially change its predictive performance, see Figure S1.

### 2.6.2    Additive model

To maximize predictive performance while retaining interpretable feature effects we developed an additive model (AM) (Buja et al., 1989; Hastie and Tibshirani, 1990). This assumes that the mean of the log-transformed MSA $Y$ is linked to the features by smooth (non-linear) functions. As these functions are unknown, we approximate them by linear combinations of user-specified basis functions. To this end, we used the standard cubic $B$-splines as bases (de Boor, 2001). The $i^{\text{th}}$ aggregated value of the $k^{\text{th}}$ feature is denoted by $x_{i,k}$, for $k = 1, 2, \ldots, K$, where $K$ is the number of features used in the model (the maximum

being $K = 80$ for Group A and $K = 155$ for Group A+B). The cubic $B$-spline basis function is generically written as $B()$. The AM main equation can be expressed according to Eq. (4):

$$Y_i = \alpha_0 + \sum_{k=1}^{K} \sum_{j=1}^{J} \alpha_{j,k} B_j(x_{i,k}) + \varepsilon_i \tag{4}$$

where $\alpha_0$ is an intercept, $J$ is the number of spline bases we use for every feature effect represented by the linear combination $\sum_{j=1}^{J} \alpha_{j,k} B_j(x_{i,k})$, the $\alpha_{j,k}$s are coefficients weighting the different splines bases for the $k^{\text{th}}$ feature effect, and $\varepsilon_i$ is an independent error term assumed to have mean zero and constant variance. To reduce the computational cost and as an indirect regularization (see below) we set $J = 5$ throughout. This implies that the spline function relies on $J - 2 = 3$ knots; these were set as the minimum, median, and maximum observed values for each feature. There are thus $P = K(J - 1) + 1$ free model

parameters. These were estimated on the training data by minimizing the mean squared error (Eq. 1). The mean squared error loss function relies on the assumed independence between the values of $\varepsilon_i$. Even though the MSA measurements were recorded sequentially in time, with the possibility of temporal dependence (autocorrelation), we believe the independence assumption is tenable here. The rationale is that if the $K$ features include a subset of relevant variables that explain and predict $Y$, then all that remains is indeed white noise represented by $\varepsilon$. In other words, we assume any (marginal) temporal dependence in $Y$ is

captured by the effect of the available features.

The main challenge when fitting such a model is that $K$ can be potentially large, leading the number of parameters $P$ to exceed the number of observations $N$. That is, the AM can easily overfit the training data, with estimated feature effects appearing overly complex (i.e., wiggly) and difficult to interpret. As we want the model to predict well out-of-sample





observations, some regularization is required. Typical regularization approaches allow for a large $J$ and involve adding

penalties to the mean squared error loss so that many values of $\alpha_{j,k}$ are shrunk towards zero or even exactly set to zero (Wood, 2017). We explored such approaches, notably using effect-specific ridge penalties or a group lasso penalty to select features as part of model fitting, but could not obtain satisfying results. These also came with undue computational overhead involved in part in selecting the penalty/smoothness hyperparameters. We thus opted for a simpler strategy: set $J = 5$, which is rather small and guarantees on its own that the estimated feature effects are relatively smooth albeit flexible enough. Rather than

enforcing some penalty to counteract a large $K$, we selected features with a forward stepwise selection (FSS) procedure (Trevor Hastie et al., 2020). This scheme starts with an empty model, only with the intercept $\alpha_0$, and sequentially adds features based on an objective criterion. Our criterion here is the prediction MSE based on the temporal CV described in the previous section. At each FSS step, the feature that reduces this CV-based MSE the most is selected and kept in the model in subsequent steps. The scheme ends when the MSE reduction is smaller than a threshold of 5% of the initial reduction from an empty model to a

model with one feature. That way, the model never includes too many variables, $P$ remains low relative to $N$, and we have the guarantee that the selected features are useful in predicting/forecasting MSA observations. This also comes with computational gains, since the independent fits at each step (one for each candidate feature) can be parallelized. After this FSS round, we explored if any pairwise interaction (product of two features) between the selected features was worth including. For this, we applied the FSS in a similar fashion and only kept the most useful interactions with the same 5% MSE reduction threshold.

500         In addition to predictions, the AM yields interpretable effects as output. After training, the estimated effect of feature $k$ on the response is calculated similarly to the mean prediction $\hat{y}_i$ presented above, where all features except the $k$th are set to their mean observed value. Therefore, only the marginal contribution of the $k$th feature remains, and this can be represented as a curve, typically represented over a scatter plot of the response plotted against the $k$th feature. We refer to these curves (and the plots by extension) as "partial effects". These partial effects can also be constructed for pairwise interactions. In this case,

the interaction between features $k$ and $l$ is computed as the mean prediction where all other features except $k$ and $l$ are set to their mean. The interaction partial effect plot is then a three-dimensional surface represented as a function of features $k$ and $l$. The partial effects were calculated using only the training set.

## 2.7  Numerical Model Output

### 2.7.1    GEOS-Chem

Output from the global chemical transport model, GEOS-Chem (v12.9.3: https://zenodo.org/records/3974569), for atmospheric concentrations of $MSA_p$ for the years 2016 and 2017 and was used in this study. Transport processes and cloud properties are driven by NASA MERRA-2 (Modern-Era Retrospective Reanalysis for Research and Applications, Version 2) reanalysis meteorology (Gelaro et al., 2017), which has a horizontal resolution of $0.5° \times 0.625°$. GEOS-Chem has a $4° \times 5°$ horizontal resolution with 47 vertical levels. The chemical reactions were calculated every 60 min and the monthly averaged

data was produced as model output. Boundary layer $MSA_p$ is calculated from GEOS-Chem output of boundary layer height,





air density, temperature, and surface pressure. The oceanic DMS emission flux is parameterized using a sea surface temperature and wind speed-dependent gas transfer velocity (Johnson, 2010) and the climatology of seawater DMS concentrations (Lana et al., 2011; Nightingale et al., 2000). GEOS-Chem contains comprehensive $HO_x$–$NO_x$–VOC–$O_3$–halogen tropospheric oxidant chemistry including recent updates to halogen chemistry and cloud processing (Bey et al., 2001; Holmes et al., 2019;

Wang et al., 2019). In addition to the original version of GEOS-Chem v12.9.3, we used the multiphase DMS oxidation chemistry scheme recently developed by Tashmim et al. (2024), while the aqueous-phase reaction of MSA and OH was omitted due to the high uncertainty in its reaction rate (Chen et al., 2018). The wet and dry deposition schemes for aerosols and gas species are based on previous studies (Amos et al., 2012; Liu et al., 2001; Wesely, 1989).

### 2.7.2   OsloCTM3

The OsloCTM3 is an offline global three-dimensional chemistry transport model with total MSA (gaseous and particulate MSA) and output for 2008-2017 was used in this study. We opted to include this model output even though it was for total MSA as modeled $MSA_p$ in the Arctic is scarce and from measurements of gaseous and particulate MSA from the MOSAiC expedition (Boyer et al., 2023; Heutte et al., 2023; Shupe et al., 2022) the ratio of gaseous to particulate MSA in the central Arctic Ocean is approx. 0.03, thus would not likely significantly influence the results of this study. OsloCTM3 is driven by

meteorological forecast data from the European Centre for Medium-Range Weather Forecasts Integrated Forecast System (ECMWF-IFS) model with a 3-hourly temporal resolution. OsloCTM3 has a 2.25° x 2.25° horizontal resolution, 60 vertical layers, and monthly temporal resolution. The lowest layer was taken as representative of surface concentrations. OsloCTM3 consists of a tropospheric and stratospheric chemistry scheme (Søvde et al., 2012) as well as aerosol modules for sulfate, nitrate, black carbon, primary organic carbon, secondary organic aerosols, mineral dust, and sea salt (Lund et al., 2018). The

sulfur cycle chemistry scheme and aqueous-phase oxidation are described by Berglen et al. (2004). The oceanic DMS emission flux in OsloCTM3 is parameterized using wind fields from ECMWF-IFS, gas transfer velocity calculations from Nightingale, (2000), and seawater DMS concentrations from Kettle and Andreae, (2000). Aerosol removal includes dry deposition and washout by convective and large-scale rain from ECMWF-IFS.

### 2.7.3   GISS-E2.1

The NASA Goddard Institute of Space Studies (GISS-E2.1) Earth system model (ESM), GISS-E2.1, is a fully coupled ESM, for a full description of GISS-E2.1, see Kelley et al. (2020). GISS-E2.1 has a horizontal resolution of 2° x 2.5° and 40 vertical layers and produced monthly output for 2008-2017. The output of the GISS-E2.1 model used historical CEDS emissions from 2008 to 2014 and SSP2-4.5 from 2015 to 2017. The lowest layer was taken as representative of surface concentrations. The tropospheric chemistry scheme used in GISS-E2.1 (Shindell et al., 2001, 2003) includes inorganic chemistry of $O_x$, $NO_x$, $HO_x$,

CO, and organic chemistry using the CBM4 scheme (Gery et al., 1989). The meteorology was nudged to the NCEP reanalysis (Kalnay et al., 1996). The one-moment aerosol (OMA) scheme used (Bauer et al., 2020) is a mass-based scheme in which aerosols are assumed to remain externally mixed and have a prescribed and constant size distribution. The OMA scheme treats



sulfate, nitrate, ammonium, carbonaceous aerosols (including methanesulfonic acid formation), dust, and sea salt. The natural emissions of DMS are calculated interactively using prescribed and fixed maps of DMS concentration in the ocean (Im et al., 550 2021).



# 3   Results and Discussion

This section begins with an analysis of the seasonal cycles and source regions of in situ MSA$_p$ observations at the High Arctic stations for context. We then evaluate current numerical models' ability to simulate MSA$_p$ followed by a performance analysis of our data-driven models. The most relevant features selected by the models are discussed, and their effects on the AM output
of MSA$_p$ are investigated.

## 3.1   In-situ MSA observations from Arctic stations

The locations and seasonal cycles of MSA$_p$ at each of the Arctic stations are displayed in Fig. 2a and b, respectively. For all stations, MSA$_p$ is elevated beginning in April and ending in September. This period corresponds to polar day, receding sea ice, increase in atmospheric oxidants as well as phytoplankton blooms. Details about each station's seasonal cycle and source
regions are given below.

Alert, the most northern station located at 210 m asl on the Canadian Archipelago (Fig. 2a and Table 1), which is surrounded by sea ice and land, experiences air mass transport mainly from the central Arctic Ocean, Canadian Archipelago, and Greenland Sea (Sharma et al., 2012). Alert exhibits a maximum in MSA$_p$ during May (0.014 [0.011, 0.021] µg m$^{-3}$, median [25$^{th}$, 75$^{th}$ percentiles]) followed by lower levels during June and July until reaching a second smaller maximum in August
(0.009 [0.006, 0.011] µg m$^{-3}$). The maximum in May is likely due to efficient transport from regions of biologically active waters in the Northern Atlantic (Sharma et al., 2012; Xie et al., 1999) while the second maximum in August likely arises from biological emissions from regions of retreating sea ice in the Arctic Ocean (Sharma et al., 2019).

Gruvebadet, located on the coast of the Svalbard Archipelago with sea ice to the north and open ocean to the south, experiences air mass transport mainly from the Greenland and Barents Sea (Becagli et al., 2016). Gruvebadet displays the
highest MSA$_p$ concentrations of all the stations, with a maximum in May (0.022 [0.011, 0.046] µg m$^{-3}$). As the summer progresses, monthly median MSA$_p$ concentrations steadily decrease, although the 75$^{th}$ percentile does display a shoulder in July showing the increased variability of MSA$_p$ during the later summer months. The May maximum is likely related to the spring bloom in the Barents Sea and the variability in the later summer is likely biological activity in the Greenland Sea as well as differences in oceanic DMS-producing species in these regions and timing/location of sea ice retreat (Becagli et al.,
575    2019).

Pituffik/Thule, located in Northwestern Greenland at 220 m asl, experiences air mass transport almost exclusively from Baffin Bay (Becagli et al., 2016). Although located close to each other, Pituffik/Thule experiences similar levels of MSA$_p$ compared to Alert but interestingly a different seasonal cycle. From May to July, median MSA$_p$ concentrations at Pituffik/Thule plateau around 0.011 [0.007 and 0.018 µg m$^{-3}$], while Alert experiences two local maxima (May and August as
discussed above). The northern section of Baffin Bay regularly experiences the North Water (NOW) polynya, which is characterized by sea-ice-free areas and upwelling of nutrients (Tremblay et al., 2002). The NOW polynya begins to form in early spring and stays open until late July when sea ice is largely absent from the region. The timing of the NOW polynya and



the associated exposure of the underlying ocean to the atmosphere and solar radiation as well as nutrient-rich upwelling (which is crucial for DMS production) is the likely cause of the rather flat $MSA_p$ seasonal cycle at Pituffik/Thule (Becagli et al., 2016).

Utqiaġvik/Barrow, located on the shores of the Beaufort Sea in the North American Arctic experiences air mass transport from the central Arctic Ocean (Chukchi and Beaufort Seas), the Bering Sea/Strait, and surrounding continental areas (Alaska, Canada, and Russia) (Moffett et al., 2020; Quinn et al., 2002; Sharma et al., 2012). Utqiaġvik/Barrow displays a different seasonal cycle compared to the other stations (Fig. 2b), with maximum $MSA_p$ concentrations occurring in later summer. Utqiaġvik/Barrow experiences an increasing pattern in $MSA_p$ concentration from April culminating in a maximum

monthly median during August (0.012 [0.006, 0.016] µg m$^{-3}$). Interestingly, the maximum 75$^{th}$ percentile (June) at Utqiaġvik/Barrow is not concurrent with the maximum monthly median (August), which indicates higher variability in June but on average higher values during August. The low values in early spring could be due to the low amounts of biological activity in the surrounding seas (Hulswar et al., 2022; Lana et al., 2011) during this time (as opposed to the biologically active waters in the Northern Atlantic during spring), whilst the late summer peak could be due to transport from more warmer, local

waters in the Northern Pacific during August (Moffett et al., 2020; Quinn et al., 2002), which is a hotspot of DMS emission (Wang et al., 2020).

The differences between the stations could be credited to the different locations, sea ice retreat timing/location, differences in the DMS-producing communities, oxidant species and levels, precipitation patterns, and different air mass transport patterns. The differences in the seasonal cycles, environmental conditions, and circulation patterns of these

geographically dispersed measurement stations allow for an investigation and modeling of the processes unique to each station from a Pan-Arctic perspective. While much research has gone into elucidating the source regions, geographic differences, and seasonal behavior of $MSA_p$, few have investigated the environmental drivers of $MSA_p$, which is one of the goals of this study.

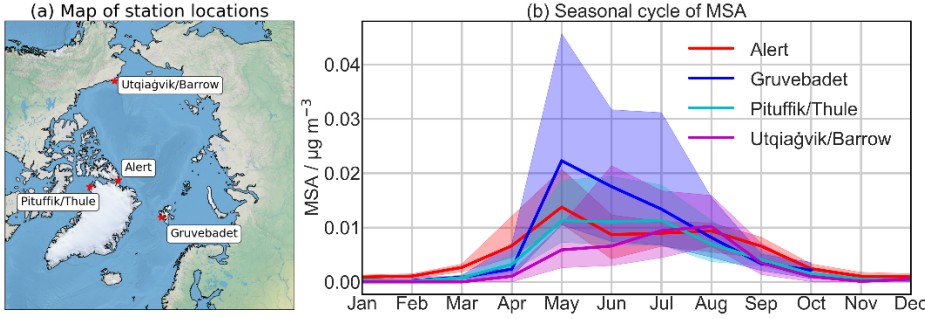


**Figure 2. Station locations and seasonal cycles.** (**a**) Map of Arctic stations marked with a red star. The map background is from Natural Earth. (**b**) $MSA_p$ seasonal cycle at Alert (red), Gruvebadet (blue), Pituffik/Thule (cyan), and Utqiaġvik/Barrow (magenta). The median is represented by the thick lines and the interquartile range is represented by the shading.



## 3.2 Comparison of numerical model output to in situ MSA concentrations

We compare in situ $MSA_p$ measurements from each Arctic station to output from three numerical models (GEOS-Chem, OsloCTM3, and GISS-E2.1) and one reanalysis product (CAMS) to gauge their current predictive abilities. Details about CAMS are given in Sect. 2.1.4 and details about the numerical models are given in Sect. 2.7. For a quantitative comparison using a regression analysis, we focus on the same evaluation metrics used for evaluating the data-driven models ($R^2$, PCC, and MSE) and limit our evaluation to the same months (April-September), we calculated the slope of predicted versus measured

$MSA_p$ as an additional metric. For a qualitative comparison, we compare the average seasonal cycles of numerical model output to in situ observations. For both the quantitative and qualitative comparison, we utilize all available years at a given station to obtain as large of sample size (and therefore a more robust statistical analysis) as possible. Our intent with such a comparison is to quantitatively gauge the current level of predictive performance for $MSA_p$ in numerical models, especially for the seasonal cycle, and for comparison against our data-driven models. We do not intend to identify and explore the

underlying causes of the discrepancies between the numerical models and observations which is beyond the scope of this work. The regression analysis and seasonal cycles of the numerical models against in-situ observations for Alert, Gruvebadet, Pituffik/Thule, and Utqiaġvik/Barrow are presented in Fig. 3, S2, S3, and S4, respectively.

Output from GEOS-Chem was only obtained for 2016-2017 therefore only a comparison at Alert, Gruvebadet, and Pituffik/Thule was possible. $MSA_p$ from GEOS-Chem is calculated over the height of the boundary layer for comparison to

observations. For all three stations, a negative $R^2$ value is observed, indicating that GEOS-Chem is worse at predicting $MSA_p$ values than the mean of the observations. PCC values range from 0.16 (Pituffik/Thule) to 0.85 (Gruvebadet), although only one year was available for comparison at Gruvebadet (Sect 2.1.1 and 2.7.1) making this result less statistically robust. MSE values range from $6.27 \times 10^{-3}$ (Alert) to $3.5 \times 10^{-2}$ µg m⁻³ (Gruvebadet) (Figs. 3, S2, and S3). Slopes larger than one are observed for all stations (ranging from 1.28 (Pituffik/Thule) to 6.67 (Gruvebadet)), indicating GEOS-Chem overestimates

$MSA_p$ relative to observations. The seasonal cycle of observed $MSA_p$ is best reproduced by GEOS-Chem at Alert, with the model able to capture the double maxima in spring and autumn (Fig. 3), although the timing and relative magnitude of the second peak in autumn are not aligned with observations.

The OsloCTM3 output is available for the entire study period, therefore, all data from all stations could be used. $MSA_p$ concentrations from the lowest model level were taken as representative of the surface level. OsloCTM3 overestimates

in situ $MSA_p$ observations at all locations, with slopes ranging from 3.5 (Pituffik/Thule) to 6.5 (Gruvebadet). Additionally, the variation and magnitude are poorly reproduced with negative $R^2$ values for all stations. The PCC slightly captures variability with values ranging from 0.18 (Utqiaġvik/Barrow) to 0.47 (Gruvebadet). MSE values range from 0.013 (Pituffik/Thule) to 0.066 µg m⁻³ (Gruvebadet). The month of peak $MSA_p$ concentrations is consistently during June in OsloCTM3 which does not reflect the variations in the timing of the seasonal maxima at the various locations. At no station does the model correctly

predict the peak month of MSA concentration.



GISS-E2.1 output is available for the entire period and the lowest model level was taken as representative of the surface. The GISS-E2.1 model generally overestimates in-situ $MSA_p$ at Gruvebadet, Pituffik/Thule, and Utqiaġvik/Barrow (slopes ranging from 1.63 to 4.2) and the observed variation is poorly captured with negative $R^2$ values and MSE values ranging from $1.78 \times 10^{-4}$ (Alert) to 0.014 µg m$^{-3}$ (Gruvebadet). At Alert, the magnitude of $MSA_p$ concentrations is best reproduced by

the GISS-E2.1 model compared to other stations as evidenced by the lowest MSE ($1.78 \times 10^{-4}$ µg m$^{-3}$), although concentrations are underestimated with a slope of 0.52 and the variation and magnitude are poorly captured with a negative $R^2$ value. PCC values range from 0.19 (Alert) to 0.64 (Gruvebadet). The peak month of $MSA_p$ concentration from the GISS-E2.1 model is consistently during June. Several features from the in situ $MSA_p$ seasonal cycles are captured by the GISS-E2.1 model, for example, the second, minor peak of $MSA_p$ during August at Alert. The peak month of $MSA_p$ concentrations at

Utqiaġvik/Barrow is August and while GISS-E2.1 does not capture this, it does show elevated levels during August. At Pituffik/Thule, the seasonal cycle is quite well captured apart from greatly overestimating concentrations during June. Overall, the GISS-E2.1 model reproduces $MSA_p$ concentrations at similar magnitudes as in-situ observations and can capture certain features of the observed seasonal cycle, although it incorrectly predicts the timing and concentrations during the peak month of $MSA_p$ levels.

The CAMS $MSA_p$ data were averaged using the median according to the start and stop time of filter samples for the respective stations. CAMS output generally, but only slightly, underestimates in situ $MSA_p$ observations with slopes for all stations ranging from 0.45 to 0.80. The variability and magnitude are poorly captured with negative $R^2$ values for all stations. The PCC is consistent for each station with values between 0.3 to 0.4 and MSE values range from $2.1 \times 10^{-4}$ (Pituffik/Thule) to $1.46 \times 10^{-3}$ µg m$^{-3}$ (Gruvebadet). The absolute values of the seasonal cycle are close to observed values, although, the peak

MSA month is incorrectly predicted by CAMS at each station. A slight shoulder is observed during May for CAMS $MSA_p$ at Alert, however, no other noticeable features of the in-situ seasonal cycle are reproduced. Overall, the CAMS reanalysis product most accurately reproduces the levels, seasonal cycle, and spatial distribution of $MSA_p$ in the Arctic, although it does not reproduce the timing of peak MSA concentrations.

In summary, we find that, in general, numerical models struggle to accurately reproduce the variability, magnitude,

and seasonal cycles of in situ $MSA_p$ observations. GEOS-Chem, GISS-E2.1, and OsloCTM3 overestimate $MSA_p$ levels and miss the timing of peak MSA concentrations. CAMS is generally able to reproduce $MSA_p$ levels with a similar magnitude compared to observations although the seasonal cycle is usually inconsistent. Although CAMS was able to most accurately reproduce the behavior of $MSA_p$, it will not be able to predict long-term future concentrations for climate analysis, being a reanalysis product capable of only short-term forecasting. Therefore, our science community still lacks the appropriate

modeling tools to accurately explore the climatic importance and future changes of $MSA_p$.

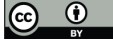



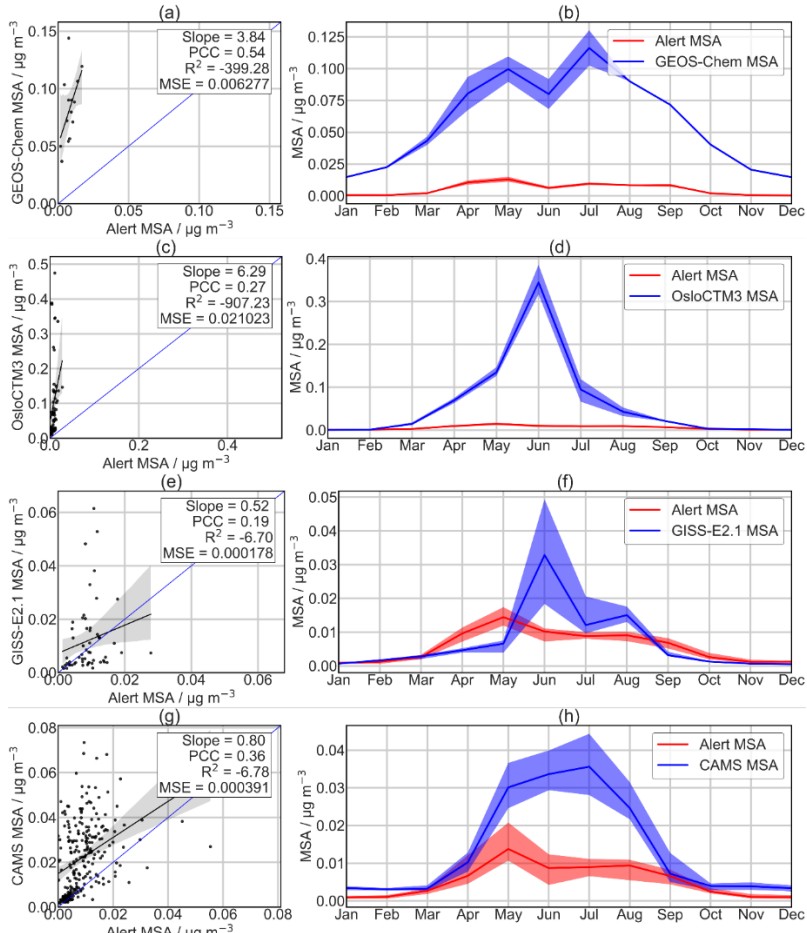

**Figure 3. Comparison of modeled against in situ MSA$_p$ observations from Alert.** Scatterplots on the left compare only April to September (over the available period for each station) with the 1:1 line in blue, linear fit in black, 95% confidence intervals estimated through bootstrapping in the shading. and and seasonal cycles on the right (thick line is the median and shading is interquartile range) for GEOS-Chem (**a** and **b**), OsloCTM3 (**c** and **d**), GISS-E2.1 (**e** and **f**), and CAMS (**g** and **h**). The MSE, R$^2$, and PCC values are calculated according to Eqs. (1), (2), and (3), respectively.

### 3.3 Data-driven model performance

In this section, we present and discuss the implemented data-driven models used to estimate ambient MSA$_p$ concentrations. We use the RF as a baseline model and focus on AM as a tailored model developed for the task at hand. Figure 4 summarizes the prediction performance in the temporal CV scheme and on the test set (Table 3) for the RF and AM with Group A+B on the four stations. The $R^2$, PCC, and MSE metrics are computed on the MSA$_p$ original scale in Fig. 4a and c, and on the log scale in Fig. 4b and d, respectively.





685        Prediction performance is relatively good on the log scale, with $R^2$ values up to 0.49 and 0.54 and PCC up to 0.74 and 0.82 for the temporal CV and test datasets respectively. Comparing the two models, AM has systematically higher CV $R^2$ (correspondingly lower MSE and similar PCC) in the St evaluations. This is expected since its variable selection procedure was designed to minimize the CV-based MSE. In the AS and ASF evaluations, neither model seems to clearly outperform the other. The $R^2$ values on the original $MSA_p$ concentration scale are lower than on the log-transformed data, with a maximum

of 0.37 and 0.29 for the temporal CV and test datasets, respectively. A likely explanation for the better performance on the log-scale could be the inter-annual, short-term variations in $MSA_p$ concentrations, which tend to be underpredicted by the models, particularly affecting the original scale data (Figs. 5 and S6), but less so for the log-transformed data which the models were trained on. The underprediction of $MSA_p$ peaks is particularly noticeable for Gruvebadet, where $R^2$ values on the log-transformed data are much higher than for the original data (Fig. 3c and d). Scatterplots and regression lines of the measured

versus modeled $MSA_p$ are displayed in Fig. S7. The regression lines for RF and AM against observations often overlap or have similar slopes, but with a slight vertical shift particularly evident for Utqiaġvik/Barrow, indicating that different models are producing different amounts of background $MSA_p$ for this station. Comparing the left side of Fig. S7 with the right side, the log transformation clearly facilitates model fitting as mentioned above, especially for Gruvebadet (Fig. S7b and f).

        Our two data-driven models are relatively complex and rely on a large number of features for this prediction task.

However, the results suggest that our models might be missing important variables or critical relationships that are not captured either due to inaccuracies in the original datasets (ERA5, CAMS, FLEXPART, etc.) or an effect of the feature engineering (averaging over daily intervals smooths out short-term temporal/spatial variation or important processes are occurring on timescales further backward in time than five days). In addition, interannual variability can cause seasons in some years to be markedly different than in other years, making the out-of-sample prediction quite challenging for low-time resolution datasets

of 8 years. This is exacerbated by splitting the dataset into training and test sets which further reduces the amount of available data for the algorithm to learn from the data, although this is an essential step in data-driven modeling. The best MSE values on the original data scale are found with the AM for Alert and Pituffik/Thule, whereas the results on the log scale are clearly best at Gruvebadet (Fig. 4c and d). The better performance for Gruvebadet, with a daily temporal resolution, can likely be explained by its training sample size ($N = 937$) being roughly three to six times larger than that of the three other stations,

highlighting the importance of high temporally resolved data. On the original MSA scale, Alert shows the lowest prediction performance, with Utqiaġvik/Barrow being a close second. Alert, with weekly temporal resolution, has the smallest training sample size ($N = 166$), again hinting at the importance of having enough observations to achieve better prediction. The modeled $MSA_p$ from RF and AM show similar temporal patterns relative to the observations for the test set years (Fig. 5), although capturing both the timing and magnitude of peaks and troughs is difficult and often only one of the two is captured at a time

(i.e., either the magnitude or timing is predicted correctly but not both).





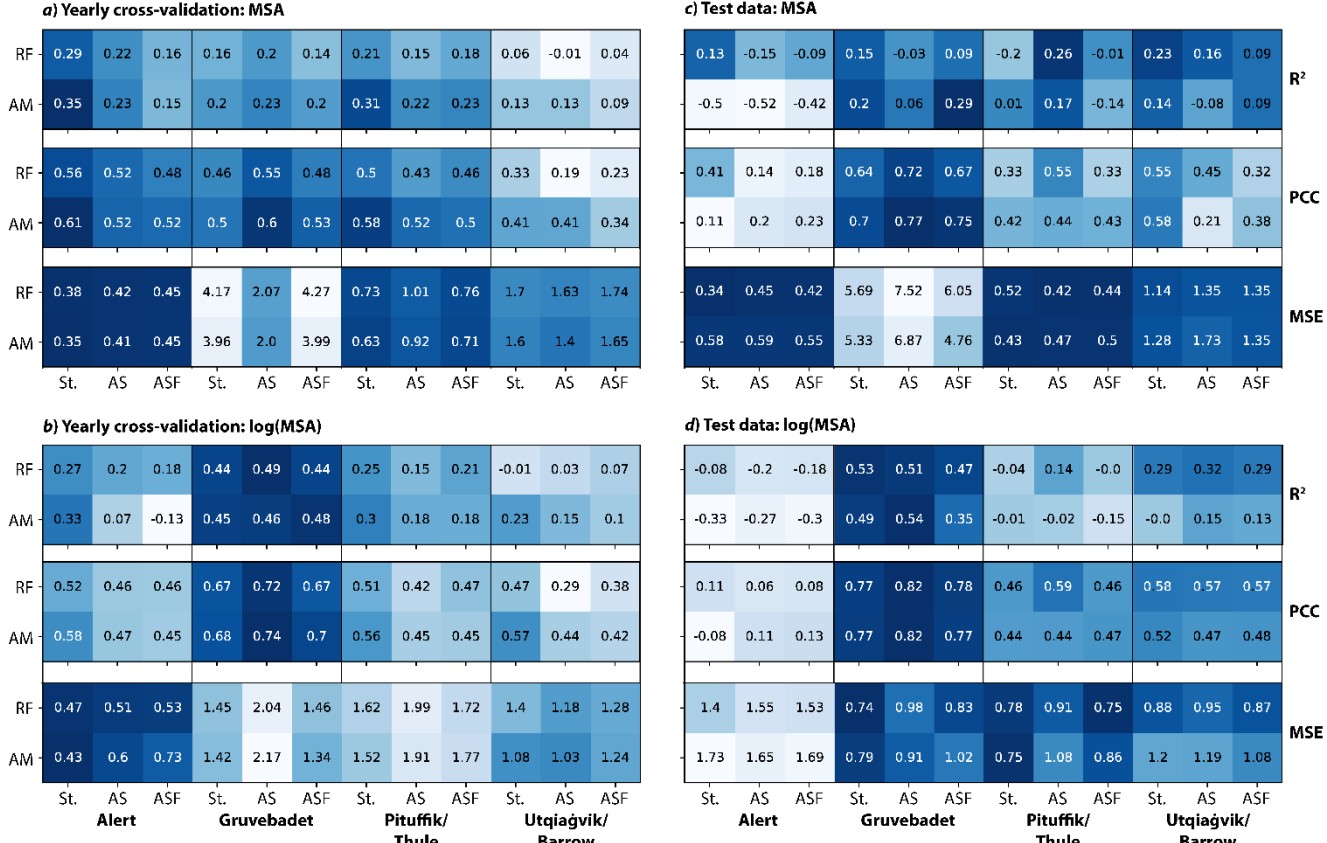

**Figure 4: Prediction performance for the temporal cross-validation (CV) scheme and on the test set for the four stations, using the selected features from Group A+B for the random forest (RF) and additive model (AM). (a)** and **(b)** show CV performance on original and log scales, respectively. **(c)** and **(d)** show performance on the test set on original and log scales, respectively. In each panel, $R^2$ is shown in the top sub-panel, the Pearson correlation coefficient (PCC) in the middle sub-panel, and the mean squared error (MSE) at the bottom. St refers to a model trained and tested on the specified station, AS refers to a subset of the data with an equal number of observations from each station, and ASF refers to all data from all four stations and tested only on the specified station. MSE is multiplied by $10^4$ to display three significant digits. The color scale indicates performance, where the darkest blue signifies the best performance (lowest MSE, highest $R^2$, and highest PCC within each row). The MSE, $R^2$, and PCC values are calculated according to Eqs. (1), (2), and (3), respectively.

Importantly, by comparing the St and ASF fits for both models, it seems that the ASF fitted values tend to have a higher spread (higher MSE, Fig. 4). That is, pooling all four stations together for a single Pan-Arctic model often yields more variable predictions, and thus rarely improves the fit locally. These geographically dispersed stations with varying seasonal cycles (Fig. 2) should theoretically allow the modeling of $MSA_p$ from a Pan-Arctic perspective (i.e., modeling processes occurring throughout the Arctic and not only at a specific station). However, the time series from the individual stations might



behave differently enough that pooling all observations together does not allow for improved modeling. The fact that models trained and tested on individual stations do not show particularly high evaluation metrics either (St. in Fig. 4) could also

contribute to this observation. The chemical and physical processes of $MSA_p$ production are necessarily similar across the Arctic, however, the relative importance of certain processes might change depending on time and location. If a station-specific model cannot capture the relationships in the data, either due to missing input variables, inaccuracies in the original input datasets used for feature engineering, inter-annual variability, or the low time resolution, then these errors will propagate into the AS and ASF datasets. These compound errors may in effect prevent the model from capturing these processes. Pooling

several geographic locations into a single data-driven model is common in ML and has been shown to provide promising results (Bertrand et al., 2023; Mansour et al., 2023; McNabb and Tortell, 2022; Zhou et al., 2023). Here our results suggest this likely only has an advantage if the individual stations can be accurately modeled.

While our data-driven models struggle to reproduce the observed $MSA_p$ with particularly high accuracy (max $R^2$ = 0.29), they can capture variability (PCC up to 0.77), and they outperform the classic numerical models. This is evident from a

comparison of the negative $R^2$ values for the numerical models (indicating the numerical models are worse at predicting $MSA_p$ compared to the mean of the observations) versus the data-driven models (Fig. 4). This shows that data-driven modeling (as opposed to the numerical modeling) has the potential to more accurately represent ambient $MSA_p$ concentrations when only considering the input data and there is still significant progress to be made in modeling natural, biogenic Arctic aerosols from a numerical and data-driven perspective. A comparison of the seasonal cycle from numerical vs data-driven models for only

the test set years (thus a direct comparison of the same periods) is shown in Fig. 9 and evaluation metrics are showed in Fig. 10. This shows our data-driven models can reproduce the seasonal cycle with greater accuracy compared to data-driven models.





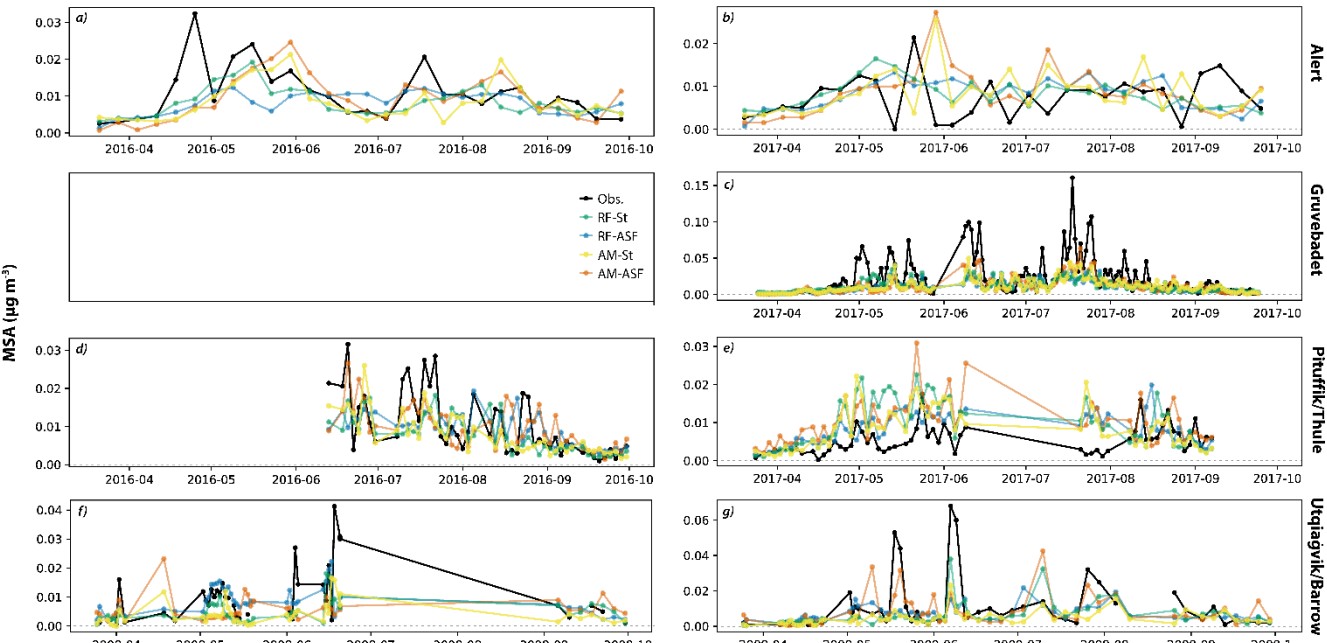

**Figure 5: Observed and modeled time series of MSA$_p$ for the test dataset at all four stations.** (**a**) and (**b**) Alert, (**c**) Gruvebadet, (**d**) and (**e**) Pituffik/Thule, and (**f**) and (**g**) Utqiaġvik/Barrow. St refers to a model trained and tested on the specified station and ASF refers to all data from all four stations and tested only on the specified station. The observations are shown in black. Data from Gruvebadet during 2016 is not available.

### 3.4 Selected features

Features contributing significantly to the RF and AM model outputs for different backward timesteps were selected from the Group A and A+B subsets for each model using the FSS (see Methods for more details). Group A included reliable features for prediction of MSA$_p$ and Group B included features expected to be good predictors of MSA$_p$, although less reliable. The right-hand panel in Fig. 6 summarizes which features are selected by which model over all timesteps, for both the Group A and Group A+B subsets of variables, for every station as well as the two additional merged datasets AS and ASF. Generally, AM selects fewer features than RF over the four stations (Table 4). AM selects between two and eight variables for Group A and between three and six for Group A+B, whereas RF selects between 14 and 44 features for Group A (an exception being Pituffik/Thule, for which the models select at most five variables in Group A, see below) and between 14 and 17 features for Group A+B. This suggests that the variables in Group A+B can explain the MSA variance using fewer variables.

The differences in selected variable counts between RF and AM can likely be explained by the fact that RF has some difficulty distinguishing between features computed at various timesteps backward for the same feature type, in comparison to AM. This is because each of the features computed for a given backward timestep tends to correlate substantially (e.g.,



meteorological conditions are usually correlated to the previous days' conditions), which can make RF feature ranking inconsistent across the different decision trees. Therefore, each decision tree will likely only select one specific timestep of a

feature, if that feature group happens to be important for $MSA_p$ overall. Thus, by averaging over all trees, the different timesteps of a given feature type are likely to be ranked similarly and the strength of the ranking score is averaged out. In contrast, AM is not an ensemble and its variable selection operates sequentially, therefore if a backward timestep for a given feature (among the five timesteps) is already included and if the other four are strongly dependent and not adding additional information to the model then they will likely not be selected. Therefore, the most relevant timestep is selected consistently with AM while

RF selects different timesteps of the same variable. Another contributing factor to the difference between the number of features selected by each model could be the sensitivity of the cutoff threshold (5 %) in the FSS procedure, which would disproportionately impact the ensemble RF model over AM. The prediction performance is similar for both models (Fig. 4), therefore we can compare the selected features for each model on an equal footing. The two models were also compared with features grouped for all five backward time steps (left-hand panel of Fig. 6), which shows that a similar number of features

were chosen for RF and AM when the backward time steps were not considered separately. While the models disagree on the number and which backward time step is important for $MSA_p$ prediction, importantly, they do agree on which features are most important, indicating these models can learn the same underlying factors that drive $MSA_p$ levels.

The features selected by each model and station combination for Group A+B are listed in Table 4, where a common theme for the type of features selected emerges. Each model and station combination tend to select a source-related feature

(either related to marine biogenic emissions, total DMS emitted or ChlA, or air mass contact with surface environments, time spent over open water, OPEN_WATER, residence time in the boundary layer, BL_RT), a chemical processing-related feature (solar radiation (SSRD), OH, $O_3$, specific humidity (Q), cloud liquid water content (LWC)), and a removal-related feature (large-scale rain rate, LSRR). For instance, AM for Gruvebadet selected four features, which are related to marine emissions of DMS (ChlA_1.2 and DMS_4.5), oxidation of DMS and its intermediates to MSA (OH_BL_0.1), and removal (LSRR_1.2).

There are, however, exceptions to this tendency, notably a removal-related feature is mainly absent from the model/station combinations (Alert RF, Gruvebadet RF, Pituffik/Thule RF/AM, and AS RF) and for Utqiaġvik/Barrow AM and RF, a source-related features are absent.

Another important observation from the analysis of the selected features is that models trained on Group A+B tend to select much fewer meteorological features than models trained on only Group A. For example, specific humidity (Q) and

temperature (T) are often selected if the smaller Group A is being used but are almost never selected when using Group A+B. A possible explanation for this is that some features that are in Group B but not in Group A correlate with such meteorological variables, likely because they are driven by or co-vary with meteorological processes, e.g., solar radiation being a proxy for OH levels. This suggests that the smaller number of features selected from Group A+B (including oceanic biological, oxidant, and precipitation features, Tables 2 and 4) are better suited to capture the variability in MSA compared to a larger number of

mainly meteorological features selected from Group A. We separated the input features into two groups to examine how the data-driven models predict $MSA_p$ when only using reliable meteorological features, and when additional chemical and oceanic



related features were used as input. Comparing the evaluation metrics between models trained on Group A variables (Fig. S5) and Group A+B (Fig. 4), we can see there are no clear systematic differences between station/model combinations trained on different input data groups. Models trained only on reliable features (Group A) can perform similarly to models trained on all features (Group A+B), therefore modeling $MSA_p$ in the Arctic can likely be achieved only using meteorological features that act as proxies for chemical and oceanic processes without negatively compromising model performance.

### 3.4.1 Source-related features

For the source-related feature type, RF and AM do not agree on the selection of DMS. RF never selects DMS while AM selects it for all sites except Pituffik/Thule. A possible explanation for this is that ChlA acts as a proxy for the biological activity that drives seawater DMS production and emission (Mansour et al., 2020; Rinaldi et al., 2013). Indeed, ChlA is chosen by RF for Gruvebadet, AS, and ASF. Importantly, AM never selects the 0–1 day back version of DMS and the earliest timestep selected is 2-3 days back for ASF as well as the 3-4 days back version for Utqiaġvik/Barrow and Pituffik/Thule. Conversely, both AM and RF select early timesteps of ChlA, with the latest being 2-3 days back. This could be due to differences in the nature of the data source, with ChlA being a satellite product vs DMS emissions being parameterized based on wind speed, sea surface temperature, and seawater DMS climatologies (see Methods). The presence of clouds, which obscure the satellite view, could also affect the timesteps selected for ChlA. Even though the ChlA dataset used should minimize the effect of clouds, their influence is still present, while the DMS climatology is unaffected by their presence. Missing ChlA was imputed and this could also affect the results shown here. The other source-related feature selected is the time air masses spent within the boundary layer and over open water (sea ice < 20 %, OPEN_WATER), with both models selecting this feature for different stations and days backward. RF selected OPEN_WATER_3.4 for Alert, while AM selected the 0-1 and 4-5 days back versions for ASF and Pituffik/Thule, respectively (Table 4). Overall, while there is some disagreement between RF and AM on which source-related features are selected, both models can learn that a certain time lag seems necessary for air mass contact with biologically active marine environments to predict $MSA_p$ well, which indicates the results from the models are physically plausible.

### 3.4.2 Chemical processing-related features

For the chemical processing-related feature type, surface shortwave radiation downwards (SSRD) is commonly selected by all models when training models on Group A features only. When training models on Group A+B features, RF also always selects at least one version of SSRD while AM only selects SSRD for Alert and Pituffik/Thule. Thus, SSRD generally appears to be a strong predictor of $MSA_p$, which is expected given the need for solar radiation in the generation of photochemical oxidants required for $MSA_p$ production, both in the gas- and aqueous-phases (Jiang et al., 2023; Wollesen de Jonge et al., 2021). AM almost exclusively selects the 0–1 timestep of SSRD, which hints at the near-immediacy of a causal relation between solar radiation, and $MSA_p$ generation, likely through the production of OH radicals and other photochemical oxidants (e.g., BrO and aqueous-phase $O_3$). Gas-phase OH radical mixing ratios (either for the boundary layer or the free troposphere) are directly selected by both models at all sites except Utqiaġvik/Barrow. When AM selects OH, it is mainly the 0–1 or 1-2 days back



timestep, and either the BL or FT versions are selected depending on the station. This indicates that OH mixing ratios are
making the largest impact on the model both aloft and close to the surface in the preceding 2 days before measurement. The
lifetime of DMS is estimated to be on the order of days in the Arctic (Breider et al., 2010; Lundén et al., 2007) although the
lifetime of the intermediate compounds dimethylsulfoxide (DMSO) and methanesulfinic acid (MSIA) are both less than one
day (Hoffmann et al., 2016; Zhu et al., 2003). This indicates that the detected $MSA_p$ could be formed in close proximity to the
measurement stations when sufficient solar radiation and photochemical oxidants become readily available (Collins et al.,
2017; Jiang et al., 2023). Interestingly, neither model selected any version of OH for Utqiaġvik/Barrow (Table 4), instead
specific humidity (Q) and cloud liquid water content (LWC) was selected and Utqiaġvik/Barrow is the only station where gas-
phase $O_3$ was selected (RF). It should be noted that gas-phase OH and $O_3$ will dissolve into the aqueous-phase, thus also
affecting aqueous-phase reactions as well. The selection of SSRD and OH at Alert, Gruvebadet, and Pituffik/Thule as well as
the selection of LWC and Q at Utqiaġvik/Barrow hints at differences between the chemical processing between these stations
during months of peak concentration. Utqiaġvik/Barrow, with its $MSA_p$ seasonal cycle peaking in late summer (Fig. 2b), is
located in the Pacific sector of the Arctic while the other stations, with $MSA_p$ peaking in early summer, are located in the
Atlantic sector (Fig. 2a). The selection of different chemical processing-related features for Utqiaġvik/Barrow and the
geographic differences in relation to biologically active waters, sea ice, and ocean dynamics could explain the different
seasonal cycle observed at Utqiaġvik/Barrow compared to the other stations. This analysis cannot quantitatively determine the
relative importance of gas- vs aqueous-phase oxidation, previous research indicates that both are likely contributing to Arctic
$MSA_p$ production (Chen et al., 2023; Kecorius et al., 2023; Pernov et al., 2024b; Shen et al., 2022). This study suggests that
depending on the time of year and geographic location, different chemical processing mechanisms might be relatively more
important. While there is disagreement between the most frequently selected timestep for DMS (4-5 days back) and ChlA (2-
3 days back), the selected timesteps for these features still mainly occur temporally before SSRD or OH when these features
are selected together, indicating that our data-driven models can learn the temporal dependencies of the source- and chemical
processing-related feature types affecting $MSA_p$.

### 3.4.3 Removal-related features

LSRR (large-scale rain rate, Table 4) was selected by most model/station combinations to represent the removal of aerosols.
Interestingly, the only other removal-related feature (time air masses experienced precipitation, PRECIP, including rain, snow,
and a mix of both) is never selected by any model/station combination (Fig. 6 and Table 4). Particulate mass quickly decreases
with initial increases in accumulated precipitation during air mass transport and levels off with larger amounts of precipitation
(Isokääntä et al., 2022; Tunved et al., 2013). The PRECIP feature only estimates the time air masses experienced precipitation
and does not account for the intensity. This could explain the selection of LSRR over PRECIP and suggests that the time air
masses experienced precipitation (regardless of type – rain, snow, or mix) is less important compared to the intensity of
precipitation (estimated by LSRR). The LSRR timesteps selected, however, showed no consistent pattern with different daily
intervals being selected for different model/station combinations (Table 4). Precipitation can have dual effects on $MSA_p$, where



precipitation closer to the station can act to remove aerosols resulting in lower MSA$_p$ while precipitation further back along the trajectory can create conditions conducive for secondary aerosol formation (for which MSA is an important component) (Khadir et al., 2023; Tunved et al., 2013; Xavier et al., 2022). These dual effects could complicate the consistent selection of

timesteps for LSRR. The below-cloud scavenging coefficient of aerosol particles reaches a minimum in the accumulation mode (Andronache, 2003), which is where MSA$_p$ mainly resides (Kerminen et al., 1997), these aspects could also complicate the selection of removal related features. Overall, this shows that the data-driven models can discern removal mechanisms for MSA$_p$, although does not specify when precipitation is important, and suggests that precipitation intensity (LSRR) is relatively more important than the total time air masses experienced precipitation (PRECIP).

### 3.4.4   Physical meteorology related features

Other feature types borne out of the FSS procedure include physical meteorology related features (e.g., boundary layer height (BLH) and wind speed (WS)) which can affect the sources, oxidation, and removal of precursors and MSA$_p$ depending on the prevailing environmental conditions. High wind speed can bring nutrients to the ocean surface thus stimulating marine biological activity and enhancing the ocean-atmosphere flux of DMS (Huebert et al., 2010; Park et al., 2013) but can also

increase the oceanic mixed layer depth thus acting to delay spring phytoplankton blooms (Henson et al., 2009). Dry deposition of trace gases and aerosol particles is largely determined by turbulence which is driven by wind speed (Farmer et al., 2021) thus higher wind speeds can enhance dry deposition velocities (Mariraj Mohan, 2016) enhancing the removal of aerosols. High boundary layer heights can promote or diminish MSA burdens: high BLHs can dilute DMS in the lower atmosphere thus enhancing emissions but also diluting the oxidants and lowering the efficacy of MSA$_p$ production. High BLHs close to the

station can also dilute MSA$_p$ concentrations. While the models mainly selected source, chemical processing, and removal related features, this shows that specific meteorological conditions can also affect MSA$_p$ variability.

### 3.4.5   Vertical Origins

Certain datasets (CAMS and ERA5 on model levels, see Methods) were vertically resolved which allowed for analysis of environmental conditions near the surface (or boundary layer, BL) and aloft (or free troposphere, FT). Similar to the different

days back timesteps of a feature, AM selects only the most pertinent feature that contributes the most to the model output, as our variable selection procedure for AM performs this process sequentially by design, while RF trees select several different timesteps and vertical origins for each feature from a random subset thus might not be a globally optimal choice. This highlights the complementary nature of these two models for the feature-engineered input data – AM selects fewer features but specifically the ones that make the largest contribution to the model output while RF can broadly indicate the important features

regardless of timestep or vertical origins. While this analysis cannot quantify the relative importance of BL or FT processes to MSA production, it is worth noting that AM for Alert and Pituffik/Thule (two stations located above the sea surface, Table 1) selected OH_FT_1.2, while Gruvebadet, Utqiaġvik/Barrow, AS, and ASF selected more BL than FT features (Table 4) and OH_BL was selected at these stations except for Utqiaġvik/Barrow where Q_BL_0.1 and LWC_BL_2.3 were selected. This



suggests that the two stations located at elevation are more influenced by FT processes than the stations located close to the
surface and the Pan-Arctic merged datasets (AS and ASF).

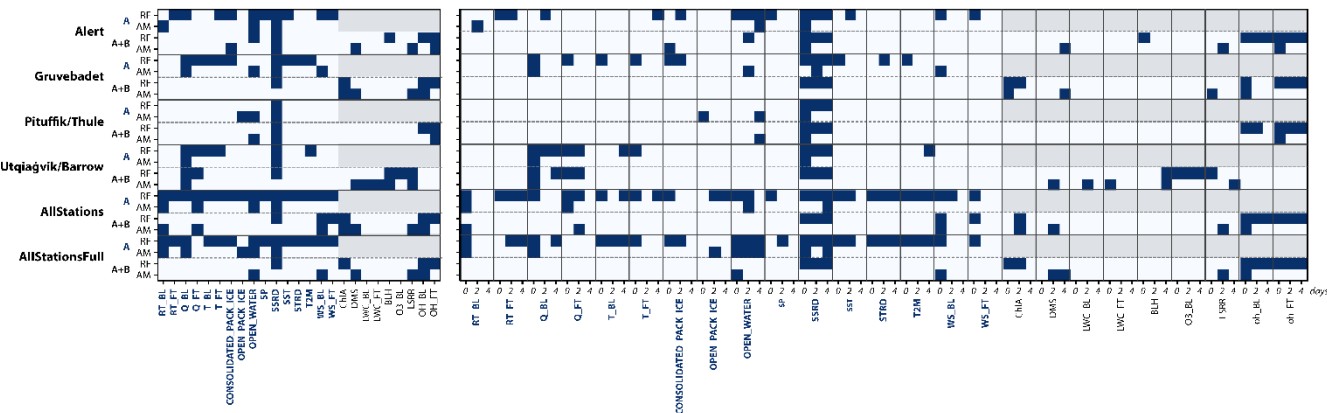

**Figure 6: Overview of features selected by the RF and AM based on Group A and Group B, by station.** The left panel
shows selected features grouped over the 0–5 days prior to each MSA$_p$ measurement, and the right panel shows the features
grouped for 0–1 (0), 2–3 (2), and 4–5 (4) days before each MSA$_p$ measurement. Features in Group A have their name in
boldface and blue type, while the additional features that are only in Group B are in regular black typeface. The grey shaded
area indicates that Group B features cannot be chosen in these model runs. St refers to a model trained and tested on the
specified station, AS refers to a subset of the data with an equal number of observations from each station, and ASF refers to
all data from all four stations and tested only on the specified station. Feature abbreviations are defined in Table 2. Only
features selected at least once by a model/station combination are presented in this figure (i.e., if a feature is not included in
the figure then it was not selected).




**Table 4: Features selected by the different models using the Group A+B set of variables.** N is the number of selected features. The chosen features are listed in order of importance for the model. Feature names are in the following format: ABBREVIATION_DAYS.BACK with the ABBREVIATION for each feature taken from Table 2 and DAYS.BACK is the daily interval backward in time preceding the measurement with the interval separated by a period and if the feature represents the boundary layer (BL) or free troposphere (FT) is also indicated, e.g., OH_BL_2.3 refers to the OH radical mixing ratio in the boundary layer 2-3 days before $MSA_p$ measurement.

| Station | Model | N | Selected Features |
|---|---|---|---|
| Alert | AM | 5 | SSRD_0.1; DMS_4.5; LSRR_3.4; CONSOLIDATED_PACK_ICE_0.1; OH_FT_1.2 |
| Alert | RF | 17 | BLH_0.1; SSRD_3.4; SSRD_4.5; SSRD_1.2; SSRD_0.1; SSRD_2.3; OH_BL_2.3; OH_BL_3.4; OH_BL_1.2; OH_FT_4.5; OH_FT_1.2; OH_BL_4.5; OPEN_WATER_3.4; OH_BL_0.1; OH_FT_0.1; OH_FT_3.4; OH_FT_2.3 |
| Gruvebadet | AM | 4 | OH_BL_0.1; ChlA_1.2; DMS_4.5; LSRR_1.2 |
| Gruvebadet | RF | 14 | OH_FT_0.1; OH_BL_0.1; OH_FT_1.2; SSRD_4.5; SSRD_3.4; OH_FT_3.4; ChlA_1.2; OH_BL_1.2; SSRD_2.3; SSRD_1.2; OH_FT_2.3; SSRD_0.1; OH_FT_4.5; ChlA_2.3 |
| Pituffik/Thule | AM | 3 | SSRD_0.1; OPEN_WATER_4.5; OH_FT_1.2 |
| Pituffik/Thule | RF | 14 | SSRD_0.1; SSRD_1.2; OH_BL_0.1; OH_FT_0.1; OH_BL_1.2; SSRD_2.3; OH_FT_3.4; OH_FT_4.5; OH_BL_2.3; OH_FT_1.2; SSRD_3.4; OH_FT_2.3; SSRD_4.5; OH_BL_3.4 |
| Utqiaġvik/Barrow | AM | 6 | Q_BL_0.1; DMS_3.4; BLH_4.5; LSRR_4.5; LWC_BL_2.3; LWC_FT_0.1 |
| Utqiaġvik/Barrow | RF | 16 | $O_3$_BL_1.2; SSRD_2.3; Q_FT_1.2; Q_FT_0.1; SSRD_4.5; LSRR_0.1; SSRD_1.2; Q_BL_0.1; BLH_4.5; Q_BL_4.5; $O_3$_BL_0.1; $O_3$_BL_2.3; $O_3$_BL_4.5; $O_3$_BL_3.4; Q_FT_3.4; SSRD_3.4 |
| AllStations | AM | 7 | OH_BL_1.2; WS_BL_0.1; DMS_3.4; LSRR_2.3; BL_RT_0.1; Q_FT_2.3; ChlA_2.3 |
| AllStations | RF | 17 | OH_FT_3.4; OH_BL_0.1; OH_FT_0.1; OH_BL_1.2; OH_FT_1.2; OH_FT_4.5; OH_BL_2.3; OH_FT_2.3; SSRD_2.3; SSRD_3.4; WS_BL_0.1; SSRD_1.2; ChlA_2.3; OH_BL_3.4; OH_BL_4.5; WS_FT_0.1; SSRD_4.5 |
| AllStationsFull | AM | 6 | OH_BL_0.1; DMS_2.3; WS_BL_0.1; LSRR_2.3; DMS_4.5; OPEN_WATER_0.1 |
| AllStationsFull | RF | 16 | OH_FT_1.2; OH_FT_0.1; OH_BL_0.1; OH_BL_1.2; OH_FT_3.4; OH_FT_2.3; OH_FT_4.5; OH_BL_2.3; ChlA_2.3; SSRD_2.3; ChlA_1.2; SSRD_3.4; SSRD_1.2; OH_BL_3.4; SSRD_4.5; OH_BL_4.5 |






**3.5 Contribution of selected features to model output (Partial effects) for Alert and Utqiaġvik/Barrow**

We investigated the relationships between the selected features and the AM output of $MSA_p$, which produces estimated partial effects (representing the contribution of a feature to the model output after accounting for all other features, see Methods for more details) for every selected feature for every station and the merged datasets (AS and ASF). Figures 7 and 8 present the

935 partial effects (as the solid red line) for the selected features at Alert and Utqiaġvik/Barrow, respectively, and the partial effects for Gruvebadet, Pituffik/Thule, AS, and ASF are displayed in Figs. S8-11, respectively. We present the partial effects for Alert and Utqiaġvik/Barrow as they are good examples of the relative importance of the two chemical processing methods observed in the study, gas- and aqueous-phase oxidation, respectively. The partial effects for each feature are discussed in order of importance from the feature selection process (Table 4). It should be noted that due to the different aggregation methods (sum

940 or mean, Table 1) over the different temporal resolutions at each station (Table 2), the magnitude and units of certain features are not comparable between stations, therefore for display purposes only the summed features were divided by average number of input data contributing to the summed feature. In this manner, the partial effects plots are comparable between stations. For each subpanel, a scatterplot of the input variables and the corresponding model output of $MSA_p$ is also included. The partial effects should not be interpreted as a fitted value of this scatterplot. The scatterplot was included to show the data distribution

945 and the low signal-to-noise ratio visible in the data: the observations have quite a large spread relative to the magnitude of the red solid curves representing the partial effects. It should also be noted that spline functions, like the B-splines used in the AM model (see Methods), are generally sensitive near the edges of the observed domain space if they contain few data points. Therefore, caution is urged when interpreting the partial effects if the data is highly skewed or if a few data points are contained at the edges of the domain space.

950 **3.5.1 Alert**

AM selected the following features at Alert, which are discussed in order of importance, SSRD_0.1, DMS_4.5, LSRR_3.4, CONSOLIDATED_PACK_ICE_0.1, and OH_FT_1.2 as well as the interactions between DMS_4.5 and SSRD_0.1 (Fig. 7).

   SSRD_0.1, a chemical processing related feature, makes a non-linear contribution to the model output of $MSA_p$, with the maximum impact on model output in a certain range of values as well as low and high values of SSRD making similar

955 contributions to model output (Fig. 7a). This indicates that there is a certain activation threshold of SSRD_0.1 required before this variable begins to increase $MSA_p$ in the model output, which is likely connected to the production of photochemical oxidants (Barnes et al., 2006; Song et al., 2022). Increasing SSRD above this threshold reduces the model output which could be due to photolysis of intermediate products during DMS oxidation or the continued oxidation of MSA to sulfate (Chen et al., 2018).

960    DMS emissions during 4-5 days prior to observation, a source related feature, shows a linearly positive relationship to model output of $MSA_p$, as expected (Fig. 7b). Although a slight change in the slope of this relationship is observed, indicating that the model output of $MSA_p$ is more sensitive to DMS emissions at lower values (with $MSA_p$ production likely being in a



DMS-limited regime) and are less sensitive at higher values (with $MSA_p$ production likely not limited by DMS availability but by other environmental conditions such as oxidants (Barnes et al., 2006)).

LSRR, a removal related feature, makes a linearly, negative contribution to the model output of $MSA_p$, indicating that precipitation acts to reduce the model output of $MSA_p$ (Fig. 7c), as expected (Isokääntä et al., 2022; Tunved et al., 2013). While this result is unsurprising, it adds validity to the model results by highlighting how they are physically interpretable.

The partial effect of CONSOLIDATED_PACK_ICE, here treated as an indirect source related feature, shows a non-linear relationship to model output of $MSA_p$ (Fig. 7d), with a maximum (minimum) at ~200 (~400) s $km^{-2}$, respectively. Alert,
being the northernmost station (Table 1) is usually surrounded by consolidated pack ice (Kwok, 2018) hence the transport time over consolidated pack ice will usually be non-zero. The maximum at ~200 s $km^{-2}$ could indicate that air masses traversed biologically productive marginal ice zones before passing over consolidated pack ice and ultimately arriving at Alert (Sharma et al., 2012). The minimum of the partial effects at ~400 s $km^{-2}$ is likely related to air masses spending time over the central Arctic Ocean and that did not come into recent contact with any major DMS source regions.

OH_FT_1.2, a chemical processing related feature, shows a non-monotonic pattern with a maximum of around ~1.5 $\times 10^{-5}$ ppbv and a minimum at ~4 $\times 10^{-5}$ ppbv (Fig. 7e). The maximum and minimum could indicate that a certain level of OH in the FT acts to produce $MSA_p$ and increasing OH above this level in the FT tends to decrease the model output of $MSA_p$. A possible explanation could be the oxidation of intermediate compounds, DMSO and MSIA, to produce $MSA_p$, and the continued oxidation of MSA to sulfate to diminish $MSA_p$ (Hoffmann et al., 2016). It should be noted that gas-phase OH will
dissolve into the aqueous phase, therefore these reactions could both occur in either phase.

Interactions between input features were also explored by multiplying the values of two input features together. Of the combinations tested for all features and stations, only the interactions between DMS_4.5 and SSRD_0.1 at Alert were retained (using the FSS with a 5% MSE reduction threshold, see Sect. 2.6.2). A contour plot of the model output of $MSA_p$ for different values of DMS_4.5 and SSRD_0.1 is shown in Fig. 7f. The results overall suggest that model output of $MSA_p$ is more
sensitive to DMS_4.5 compared to SSRD_0.1 as indicated by the higher variability of $MSA_p$ model output over the range of DMS_4.5 at a fixed value of SSRD_0.1. This is especially evident for values of SSRD_0.1 above 400 W $m^{-2}$, with a ridge of the maximum model output of $MSA_p$ for values of SSRD_0.1 around ~700 W $m^{-2}$ (Fig. 7f). Taken together, this could indicate that at Alert, $MSA_p$ production is likely limited by DMS emissions and not necessarily by the availability of solar radiation (and therefore photochemical oxidants).




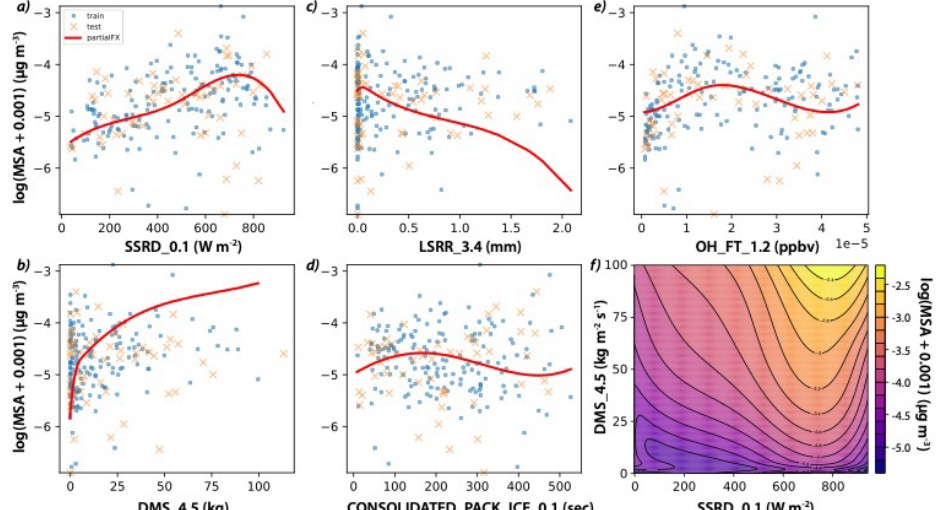

**Figure 7: AM-St partial effects for the selected features at Alert.** (**a**) SSRD_0.1, (**b**) DMS_4.5, (**c**) LSRR_3.4, (**d**) CONSOLIDATED_PACK_ICE_0.1, (**e**) OH_FT_1.2, and (**f**) DMS_4.5 and SSRD_0.1. For all panels except the bottom-right one, the red solid line is the partial effect for a different feature, blue points are the training observations, and orange crosses are the test data. The contour plot in the bottom-right panel shows the interaction effect between SSRD_0.1 and DMS_4.5, where the joint partial effect is represented by the color gradient. Feature abbreviations are defined in Table 2. St refers to models trained and tested on the specified station. Features aggregated as sums over filter time windows (see Table 2) are rescaled here by the average number of 3-hourly samples in each summation to help compare partial effects between stations.

### 3.5.2    Utqiaġvik/Barrow

For Utqiaġvik/Barrow, AM selected the following features, Q_BL_0.1, DMS_3.4, BLH_4.5, LSRR_4.5, LWC_BL_2.3, and LWC_FT_0.1.

Specific humidity is the mass of water vapor per mass of moist air and here is used as a proxy of aqueous-phase processing of DMS and its intermediates. For Q_BL_0.1, which represents the specific humidity in the boundary 0-1 days prior to measurement, the lower end of the feature range ($< \sim 0.0025$ kg kg$^{-1}$) shows a small local maximum at $\sim 0.00125$ kg kg$^{-1}$. At the upper end, a linearly, positive relationship between Q_BL_0.1 and the model output of $MSA_p$ is observed (Fig. 8a). This indicates that at lower values of Q_BL_0.1, the model output is showing a slight increase in $MSA_p$ and little variation with low values Q_BL_0.1 levels and at higher values of Q_BL_0.1 the model output of $MSA_p$ responds linearly. This relationship hints that at low values of Q_BL particles are not deliquesced yet and gas-phase oxidation could be more important while at higher values of Q_BL, sufficient aerosol liquid water is present for aqueous-phase processes to become dominant. Another explanation for this relationship could be that moist air masses arrived from within the boundary layer and from marine regions, which would carry a higher signal of moisture uptake, although no air mass history features indicating transport





from marine regions (e.g., DMS, ChlA, OPEN_WATER, RT_BL) were selected for Utqiaġvik/Barrow suggesting this is improbable.

The partial effects for DMS_3.4 display a U-shaped relationship for values < ~200 kg and afterward increase linearly (Fig. 8b). The partial effects start to decrease at high values of DMS_3.4, although the few data points in the region add uncertainty to this slope change. The majority of the data for DMS_3.4 is skewed towards lower values, which likely contributes to the U-shaped partial effects below ~200 kg. Overall, the model output of $MSA_p$ increases with increasing DMS, again showing physically plausible results.

The BLH 4-5 days prior to observation show an overall positive, linear relationship with the model output of $MSA_p$, although with some complex structure present (Fig. 8c). In the Arctic, and especially over sea ice, the BLH is largely controlled by wind shear-induced turbulence and cloud top radiative cooling (Nilsson, 1996; Overland, 1985; Tjernström et al., 2015). Recently, a gridded dataset of in situ-produced biogenic MSA (generated using machine learning) was published (Mansour et al., 2024) for the Northern Atlantic. An inverse relationship between BLH and in situ MSA was found, indicating that higher
BLHs dilute the concentrations of MSA (Mansour et al., 2024). A machine learning study on the drivers of aerosol chemical composition from Svalbard indicated an inverse relationship between BLH and biogenic-type aerosols (Song et al., 2022). These studies indicate that lower BLHs act to increase the concentration of MSA while higher BLHs dilute MSA in the lower troposphere. Our recent study analyzing the environmental drivers of MSA from a geographic perspective revealed the relationship between MSA and BLH is complex and displays different patterns in different months (Pernov et al., 2024b), with
high values of BLH tending to increase the model output of MSA in all months but low BLHs also increased modeled MSA during June and July. Our recent study and this work indicate that higher BLHs act to increase the modeled output of $MSA_p$, which could be due to higher wind speeds (and thus higher BLHs) diluting atmospheric DMS levels therefore increasing the ocean-air flux of DMS. This also highlights the differences of considering air mass history when analyzing the relationships between aerosols and environmental drivers as opposed to considering only local, in situ explanatory variables.

The partial effects for LSRR_4.5 show a somewhat unexpected relationship, with a maximum at ~1 mm and a linearly, negative relationship afterward (Fig. 8d). The minimum at > ~ 4 mm is likely highly uncertain due to the low number of data points at the end of the feature domain space. A negative relationship is expected (and observed at other stations in this study and the literature) since precipitation acts to remove aerosols. The maximum of the LSRR partial effects at a non-zero value could be related to enhanced cloudiness and thus enhanced aqueous-phase processes although unlikely since AM selected the
4-5 days back version of this feature. Another possible explanation could be that low values of precipitation 4-5 days prior to measurement act to remove particles containing a high fraction of (possibly anthropogenic) sulfate (which are acidic and hygroscopic). Depending on the acidity of the remaining aerosols, this would create conditions that would favor the selective condensation of gas-phase MSA or diminish the evaporation of aqueous-phase produced MSA in less acidic particles since MSA has been shown to selectively condense on alkaline particles (Dada et al., 2022; Yan et al., 2020). The exact cause of the
maximum LSRR_4.5 partial effect remains to be seen at this time and requires further investigation.



LWC_BL_2.3 (defined as the boundary layer cloud liquid water content 2-3 days prior to measurement) is the amount of cloud liquid water and thus an excellent proxy for aqueous-phase processing. The partial effects for LWC_BL_2.3 display two local maxima: one at ~0.5 × $10^{-5}$ and another at ~4 kg $kg^{-1}$ (Fig. 8e) albeit with an overall linearly, positive relationship with the model output of $MSA_p$. The decrease in the partial effects after ~4 kg $kg^{-1}$ carries added uncertainty due to the few

data points but could possibly suggest the effect of precipitation at high values of LWC thus acting to remove MSA. These two local maxima of LWC_BL_2.3 could indicate that gas-phase and aqueous-phase oxidation are the dominant mechanisms at lower and higher values of LWC_BL_2.3, respectively. If so, then the overall linearly, positive relationship for LWC_BL_2.3 and model output of $MSA_p$ could also indicate that aqueous-phase oxidation produces relatively greater amounts of $MSA_p$ compared to gas-phase oxidation, which is in line with the theoretical understanding (Chen et al., 2018; Hoffmann

et al., 2016).

The amount of cloud liquid water in the free troposphere 0-1 days prior to measurement (LWC_FT_0.1) shows a similar relationship to model output of $MSA_p$ as does LWC_BL_2.3, with two local maxima, an overall positive relationship, and a decrease in model output after the second local maxima (Fig. 8f). Noticeable exceptions include the overall response of model output of $MSA_p$ being less sensitive to increases in LWC_FT_0.1 compared to LWC_BL_2.3 and the decrease in model

output after the second local maxima being more substantial. This relationship likely points towards gas- and aqueous-phase oxidation occurring at differing levels of LWC but also that the model output of $MSA_p$ is less sensitive to LWC in the FT than in the boundary layer and that high values of LWC in the FT more strongly removes aerosols than in the BL.

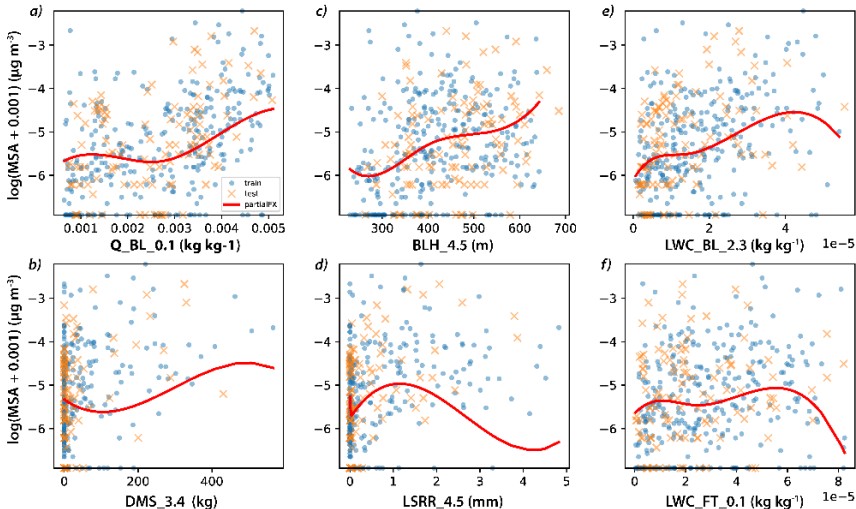

**Figure 8: AM-St partial effects for the selected features at Utqiaġvik/Barrow.** (**a**) Q_BL_0.1, (**b**) DMS_3.4, (**c**) BLH_4.5, (**d**) LSRR_4.5, (**e**) LWC_BL_2.3, and (**f**) LWC_FT_0.1.The red solid line is the partial effect for a different feature in each panel, the blue points are the training observations, and the orange crosses are the test data. Feature abbreviations are defined



in Table 2. St refers to models trained and tested on the specified station. Features aggregated as sums over filter time windows (see Table 2) are rescaled here by the average number of 3-hourly time steps to help compare partial effects between stations.


### 3.5.3    Summary of Gruvebadet, Pituffik/Thule, AS, and ASF

For Gruvebadet, AS, and ASF, a source, chemical processing, and removal related feature type were selected for AM, except for a removal related feature being selected at Pituffik/Thule. The partial effects of the selected features for AM are discussed in detail in Sect. 3 of the SI. The chemical processing-related feature type was usually OH for Alert, Gruvebadet,

Pituffik/Thule, and ASF while for Utqiaġvik/Barrow, Q and LWC were selected (Table 4) and interestingly the removal-related feature type (LSRR) showed a maximum at a non-zero value (Fig. 8d). The partial effects of Q and LWC show the presence of two local maxima and an overall positive relationship to model output of $MSA_p$ suggesting that the dual effects of gas- and aqueous-phase chemical processing can be detected. Our previous study showed the importance of both gas- and aqueous-phase oxidation for the geospatial modeling of Pan-Arctic MSA, with shortwave surface radiation (SSRD in this

study), temperature (T2M), longwave surface radiation (STRD), and low cloud cover (LCC) being the top 4 important features. Interestingly, neither T2M, STRD, nor LCC were selected for any station/model combination (Table 4). It should be noted that Pernov et al. (2024b) utilized a different feature engineering procedure to account for air mass transport patterns, a different data-driven model (gradient boosted trees vs RF/AM in this study), different explainability methods (SHAP (Lundberg and Lee, 2017) vs partial effects in this study), and focused on geospatial source regions from a Pan-Arctic perspective and not

time series of time-resolved air mass history features at individual stations, therefore a direct comparison is complicated by these facets. Although for the AS partial effects, the dual mechanisms of gas- and aqueous-phase oxidation are observed (both OH_BL_1.2 and Q_FT_2.3 were selected), indicating that modeling a merged Pan-Arctic dataset can detect these dual processes are occurring, similar to our previous research. The ASF features selected by AM and RF did not include an aqueous-phase related oxidation feature (Table 4), which could be due to Gruvebadet contributing the greatest number of samples to

the ASF merged dataset thus features selected at this station could dominate the selected features for ASF. Overall, the selected features and their partial effects for the individual stations and merged datasets show our data-driven model produces physically realistic and interpretable results.




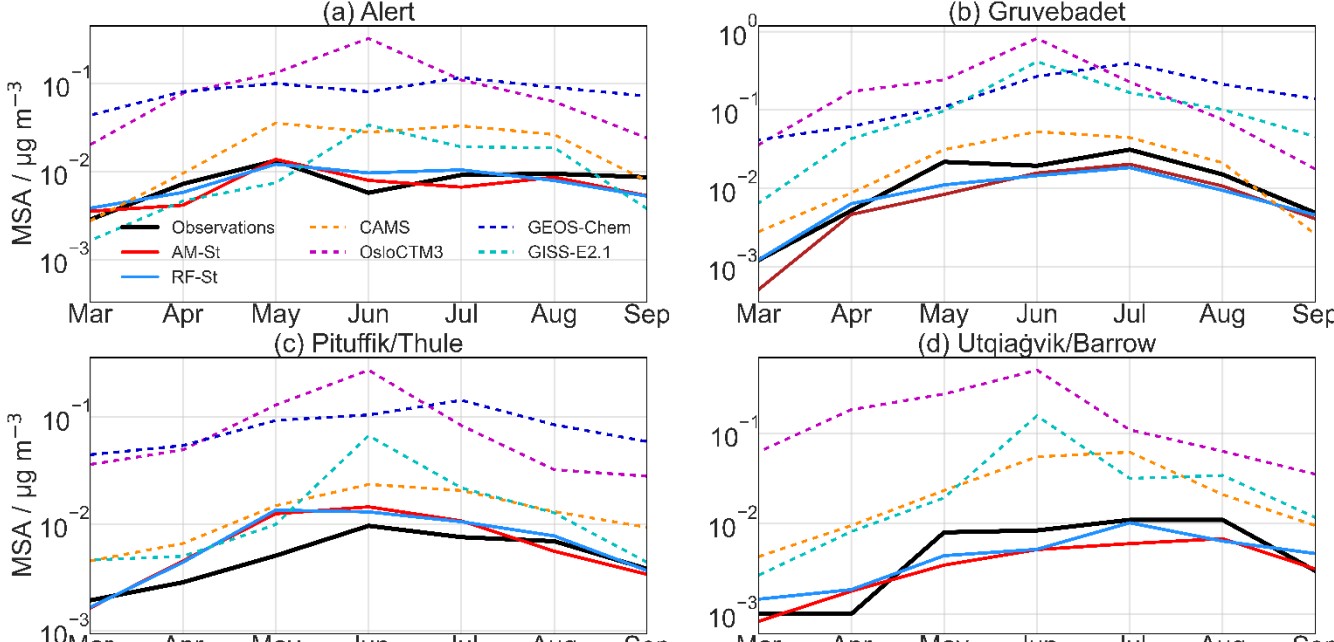

**Figure 9: Comparison of seasonal cycles for observations, St data-driven model, and numerical models. (a)** Alert **(b)** Gruvebadet, **(c)** Pituffik/Thule, and **(d)** Utqiaġvik/Barrow. Monthly medians for observations (solid black), data-driven model (AM-St in solid red and RF-St in solid light blue), CAMS (dashed orange), GEOS-Chem (dashed dark blue), GISS-E2.1 (dashed cyan), and OsloCTM3 (dashed magenta). Only data for the tests were included in this analysis for a fair comparison, see Table 3 for dates. St refers to models trained and tested on the specified station. The evaluation metrics for each data-driven and numerical model against in situ observations are given in Fig. 10.



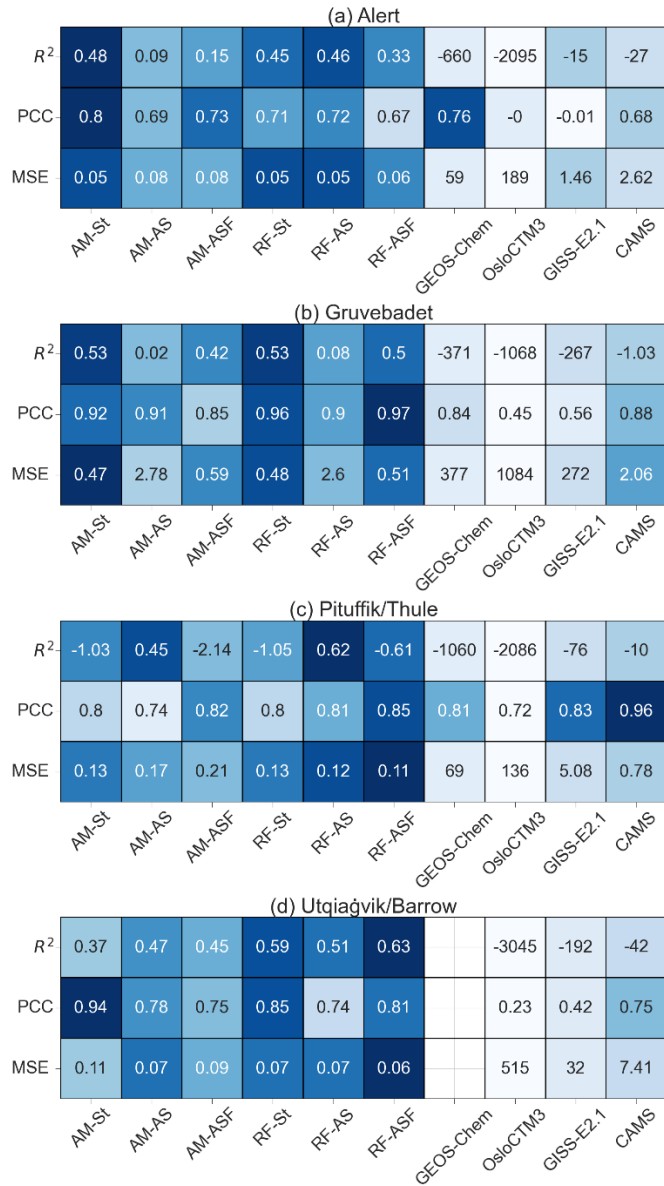

**Figure 10. Prediction performance for the data-driven and numerical models on the test set for the four stations for the random forest (RF) and additive model (AM).** (**a**) Alert (**b**) Gruvebadet, (**c**) Pituffik/Thule, and (**d**) Utqiaġvik/Barrow. In each panel, $R^2$ is shown in the top sub-panel, the Pearson correlation coefficient (PCC) in the middle sub-panel, and the mean squared error (MSE) at the bottom. St refers to a model trained and tested on the specified station, AS refers to a subset of the data with an equal number of observations from each station, and ASF refers to all data from all four stations and tested only on the specified station. MSE is multiplied by $10^4$ to display three significant digits. The color scale indicates performance, where the darkest blue signifies the best performance (lowest MSE, highest $R^2$, and highest PCC within each row). The MSE, $R^2$, and PCC values are calculated according to Eqs. (1), (2), and (3), respectively.



## 4    Conclusions

The Arctic is undergoing drastic environmental changes inducing alterations in the natural aerosol population, which in turn
affect the Arctic climate. Due to complex feedback mechanisms in the Arctic climate system, numerical modeling is vital for
understanding and predicting upcoming climate change and the role of natural aerosols therein. However, numerical models
are deeply challenged in representing natural aerosols across the Arctic. Data-driven modeling can be a faster and less
computationally intensive alternative for simulating Arctic aerosol processes, which can also identify important processes and
variables to inform improvement efforts for numerical models. Therefore, we developed an alternative data-driven modeling
approach for modeling Arctic $MSA_p$ using long-term in situ observations of $MSA_p$ from four High Arctic stations.

We developed an AM for the task of predicting $MSA_p$ observations. This tailored model allowed for more
interpretable estimated relations (partial effects) in a more parsimonious format than the RF model, which served as a baseline,
and this with both data-driven models achieving similar out-of-sample prediction performance. We incorporated feature
selection procedures into both data-driven models which selected similar features when not considering the temporal
dimension (timestep) of the features. However, RF selected more features compared to AM, when considering the timestep,
which could be attributed to the importance of different timesteps being averaged out over the ensemble of decision trees in
RF versus AM which only selected the most important timestep for each feature. We utilized two groups of features for data-
driven modeling: one with only reliable features, and one with all features related to MSA production regardless of data source
and degree of reliability. When modeling using only reliable features (which were mainly meteorological), they can act as a
proxy for unreliable features (e.g., solar radiation (SSRD) acting as a proxy for OH radical mixing ratios), although no
systematic change in model performance was detected when including all features. Indicating that a similar model performance
can be achieved by only using meteorological features but incorporating source, chemical processing, and removal related
features (albeit with added feature uncertainty) resulted in fewer features being selected.

We show that existing numerical models struggle to accurately simulate $MSA_p$ in terms of the magnitude, seasonality,
and peak months of concentrations, which can have consequences for accurate estimations of the surface energy budget and
climate projections given the role of $MSA_p$ in the climate system (Fung et al., 2022; Mahmood et al., 2019). Our data-driven
models outperform current numerical models for reproducing observations of $MSA_p$, which is especially evident for the
seasonal cycle. While data-driven models trained on merged datasets (AS and ASF) already outperform numerical models, the
accuracy achieved by training on individual stations (St), is even higher (Fig. 4). Based on the correlation of monthly medians
for the test set for each station, both the additive model (AM) and random forest (RF) can generally reproduce the seasonal
cycle of $MSA_p$ with greater accuracy than the numerical models based on the evaluation metrics used (Fig. 9, 10, S12, and
S13), although there are few exceptions depending on station, dataset, and numerical model.

Both models consistently selected features that were related to the source of MSA precursors (emission of DMS and
air mass contact with biologically productive marine areas), chemical processing of DMS (and its intermediates) to MSA
(SSRD, OH, specific humidity (Q) and cloud liquid water content (LWC)), and removal of aerosols (large-scale rain rate,





LSRR). The timesteps selected by both models indicate that they can learn the correct timing of important processes related to MSA production, for instance when DMS and SSRD were selected together, the timesteps for DMS emission always preceded those of SSRD. The features also included a vertical dimension (boundary layer vs free troposphere). Results showed that the two stations located at elevated altitudes (Alert and Pituffik/Thule) were likely more influenced by processes in the free

troposphere than in the boundary layer while the other stations (Gruvebadet and Utqiaġvik/Barrow) and merged Pan-Arctic datasets showed greater influences from the boundary layer. The relationships between the input features and the model output of $MSA_p$ were investigated through the partial effects produced by AM.

For Alert, Gruvebadet, Pituffik/Thule, and ASF, OH was the main chemical processing-related feature selected while for Utqiaġvik/Barrow, LWC and Q were selected, and for AS, both OH and Q were selected (Table 4). The selected features

for AS suggest the dual effects of gas- and aqueous-phase processing are occurring on a Pan-Arctic scale. The selected features and their partial effects for individual stations reveal site-specific characteristics that are likely contributing to the differing $MSA_p$ seasonal cycles for stations located in different sectors of the Arctic.

While our methodology can outperform current numerical models there is room for improvement. Our in-situ observations were based on long-term datasets of low temporal resolution aerosol filter samples and were therefore limited in

sample size. The input features were aggregated to the same temporal resolution as the collected filters, therefore fully capturing processes occurring on shorter time scales than the filter collection periods can be challenging. This is reflected in the ranking of data-driven model performance for the individual stations (Gruvebadet > Pituffik/Thule > Utqiaġvik/Barrow > Alert), which directly mirrors the decreasing temporal resolution of these stations (Table 1). Long-term, high temporal resolution $MSA_p$ measurements are essential for accurately capturing processes that are short-lived and highly variable.

Essential sources and sinks related to the burden of MSA, e.g., DMS emission and precipitation, while included in our model, are difficult to accurately represent using climatology-based parameterizations and reanalysis products, respectively, and improved estimations could be incorporated in future data-driven model updates. Representation of specific oxidants (e.g., halogen radicals and dissolved oxidants) is missing from our input features due to a lack of adequate datasets. Incorporating accurate representations of these crucial species would help the data-driven models elucidate the relative importance of gas-

versus aqueous-phase oxidation of DMS and specific oxidants. One of the main shortcomings of the data-driven models is the inability to capture peak or minimum concentrations, which could be due to the low temporal resolution of the input target data into the models, inadequate representation of sources/sinks, the input data missing important features (such as dissolved oxidants or halogen species), processes occurring on timescales longer than the 5 days utilized in this study, or the daily interval between timesteps being too coarse. Future data-driven modeling efforts could focus on capturing the drivers related to these

extremes of the MSA distribution.

This data-driven modeling methodology using the time-resolved air mass history can be applied to other atmospheric constituent datasets at these Arctic stations for the study period, allowing researchers to investigate other natural aerosol components or precursor species (e.g., sea salt or dimethyl sulfide) in a consistent and time efficient manner.



We recommend that numerical models be evaluated for the following processes that we identified as critical with our

two data-driven model approaches: DMS emission, chemical processing, and removal.

- Oceanic emission of DMS is the initial step for MSA formation and AM identified DMS emission, OPEN_WATER, and ChlA as key features. The numerical models in this study all utilized climatologies of seawater DMS concentrations and parameterizations for estimating the DMS flux. Updating DMS emissions schemes using data-driven modeling can help improve estimates of MSA and sulfate as well as radiative forcing (Mansour et al., 2023; McNabb and Tortell, 2022;

Regayre et al., 2020; Wang et al., 2020; Zhao et al., 2022). Current DMS emission parameterizations rely on seawater concentration, sea surface temperature, and wind speed (Johnson, 2010; Lana et al., 2011; Nightingale et al., 2000) although studies show real-world emissions are affected by atmospheric DMS levels, air temperature, pH, and nutrient availability (Hopkins et al., 2023; Kloster et al., 2007; Steiner et al., 2012; Sunda et al., 2007; Zhao et al., 2024; Zindler et al., 2014). Improved DMS emission inventories should be a focus of the modeling community going forward either

through updated parameterizations or data-driven estimates (Joge et al., 2024a, b).

- The data-driven models identified gas- and aqueous-phase oxidation to be affecting peak concentration months at different locations around the Arctic, namely OH, SSRD, LWC, and Q. Numerical models employ a plethora of chemical schemes for the oxidation of DMS and its intermediates, although shortcoming exists regarding aqueous phase oxidation, rate reaction coefficients, and oxidant concentrations (Bhatti et al., 2024; Cala et al., 2023; Chin et al., 1996; Fung et al., 2022;

Hoffmann et al., 2021; Revell et al., 2019; Tashmim et al., 2024). This and our previous work (Pernov et al., 2024b) point towards the dual effects of gas- and aqueous-phase oxidation both being key processes. Improvements to chemical processing schemes, especially aqueous-phase oxidation, as well as the inclusion of oxidants (halogens) and intermediates (DMSO, MSIA, and HPMTF) and their concentration levels should be a priority of the modeling community going forward (Chen et al., 2018; Hoffmann et al., 2021; Jongebloed et al., 2024; Tashmim et al., 2024).

- The removal of MSA through wet deposition (LSRR) was found to be a key feature identified via AM at all stations/datasets except for Pituffik/Thule and Utqiaġvik/Barrow (Table 4), however wet deposition is the key removal mechanism of MSA (Chen et al., 2018). Although dry deposition was not explicitly represented by our features, wind speed can be used as a proxy and was only selected by AM when considering the AS and ASF datasets and their relationship with MSAp output was negative (Figs. 10 and 11). Numerical models could benefit from improvements in

representations in wet deposition including aerosol activation, below- and in-cloud scavenging, and precipitation efficiency (Stier et al., 2024) as well as improvements in dry depositional processes.

Altogether, this study shows that (1) existing numerical models cannot yet simulate Arctic MSA$_p$ accurately, (2) data-driven models can outperform current numerical models although with modest performance, and (3) data-driven models can capture physically meaningful relationships between input features and MSA predictions quite well and reveal specific processes

occurring at the different stations. While data-driven modeling can aid in simulating levels of natural Arctic aerosol and provide understanding of its drivers, it struggles with extrapolating beyond the distribution space of its training dataset therefore numerical modeling is ultimately needed to predict the effects of a future climate on natural Arctic aerosol.



# 5    Appendix

**Table A1. Commonly used abbreviations**.

| | |
|---|---|
| $MSA_p$ | Particulate methanesulfonic acid |
| AM | Additive model |
| RF | Random Forest |
| BL | Boundary layer |
| FT | Free troposphere |
| St | Station specific model |
| AS | AllStations |
| ASF | AllStationsFull |
| DMS | Dimethly sulfide |
| OH | Hydroxyl radical |
| $O_3$ | Ozone |
| CCN | Cloud condensation nuclei |
| GAM | Generalized additive model |
| CV | Cross-validation |
| MSE | Mean squared error |
| PCC | Pearson correlation coefficient |
| FSS | Forward stepwise selection |




*Data availability.*

The datasets used and/or analyzed during the current study are available on reasonable request from the corresponding author Jakob Boyd Pernov (jakob.pernov@epfl.ch). ERA5 data is available from the CDS (https://cds.climate.copernicus.eu/#!/home, last accessed 08/11/2022). CAMS data is available from the ADS (https://ads.atmosphere.copernicus.eu/cdsapp#!/home, last accessed 08/11/2022). DMS emissions are available at https://ads.atmosphere.copernicus.eu/cdsapp#!/dataset/cams-global-emission-inventories?tab=overview (last access 15 September 2022). In situ MSA is available online for Utqiaġvik/Barrow (https://data.pmel.noaa.gov/pmel/erddap/tabledap/submicron.html) and Alert (https://ebas.nilu.no/) while Pituffik/Thule and Gruvebadet are available upon request. FLEXPART is available upon reasonable request to Eliza Harris (eliza.harris@sdsc.ethz.ch). Chlorophyll-a is available at https://www.globcolour.info/ (last accessed 1 October 2022).

*Code availability.*

The underlying code for this study is available as a Renkulab project (https://gitlab.renkulab.io/arcticnap/msamodeling) and by contacting the corresponding author Jakob Boyd Pernov (jakob.pernov@epfl.ch) or Michele Volpi (michele.volpi@sdsc.ethz.ch). Code for FLEXPART and python packages (xESMF, cdsapi, and RF) are available online.

*Acknowledgments.*

This research was funded by the Swiss Data Science Center project C20-01 Arctic climate change: exploring the Natural Aerosol baseline for improved model Predictions (ArcticNAP). JS holds the Ingvar Kamprad Chair for Extreme Environment Research sponsored by Ferring Pharmaceuticals. This project received funding from the Ingvar Kamprad Chair funded by Ferring Pharmaceuticals. This is a PMEL contribution 5669 and CICOES publication number 2024-1341 and was also funded by the U.S. National Science Foundation (2127733). We would like to thank Environment and Climate Change Canada, CCMR, CRD, Sangeeta Sharma, technicians (Joe Kovalik, Armand Gaudenzi, Dave Halpin, Dan Veber, D. Toom, and Alina Chivulescu), students, and operators at the Alert station who assisted in the filter collection, processing, and laboratory analyses of methanesulfonic acid. The Department of National Defense and Canadian Armed Forces staff is acknowledged for providing support at the Canadian Forces Station Alert. We would also like to thank the technicians, support staff, students, and scientists who helped obtain, analyze, and process the MSA measurements at Pituffik/Thule, Gruvebadet, and Utqiaġvik/Barrow. We would like to specifically thank Silvia Becagli and Rita Traversi from the University of Florence and the Institute of Polar Sciences at the National Research Council of Italy for providing in situ aerosol measurements from Gruvebadet and Pituffik/Thule. This research was partially funded by the Italian Ministry of University and Research (MIUR) within the framework of the projects: Dirigibile Italia: A platform for a multidisciplinary study on climatic changes in the Arctic region and their influence on temperate latitudes (PRIN-2007); ARCTICA-ARCTic Research on the Interconnections between Climate and Atmosphere (PRIN 2009); Observations of changes in chemical composition and physical properties of Polar Atmospheres from NDACCStations (PNRA 2010–2012); ARCA-Arctic: present climatic change and past extreme events



(MIUR 2014–2016); SVAAP—The study of the water vapor in the polar atmosphere (PNRA 2015–2016); OASIS-YOPP—

Observations of the Arctic Stratosphere In Support of YOPP (PNRA 2016–2018).

*Author contributions.*

Jakob Boyd Pernov – Conceptualization, Methodology, Software, Validation, Formal analysis, Investigation, Resources,
Data Curation, Writing - Original Draft, Writing - Review & Editing, Visualization, Supervision, Project administration

William H. Aeberhard – Conceptualization, Methodology, Software, Validation, Formal analysis, Investigation, Resources,
Data Curation, Writing - Original Draft, Writing - Review & Editing, Visualization, Supervision, Project administration

Michele Volpi – Conceptualization, Methodology, Software, Validation, Formal analysis, Investigation, Resources, Data
Curation, Writing - Original Draft, Writing - Review & Editing, Visualization, Supervision, Project administration

Eliza Harris – Conceptualization, Methodology, Software, Validation, Formal analysis, Investigation, Resources, Data
Curation, Writing - Original Draft, Writing - Review & Editing, Visualization, Supervision, Project administration
Benjamin Hohermuth – Conceptualization, Methodology, Software, Validation, Formal analysis, Investigation, Resources,
Data Curation, Writing - Original Draft, Writing - Review & Editing, Visualization

Ragnhild B. Skeie - Software, Validation, Resources, Data Curation, Writing - Review & Editing

Sakiko Ishino - Software, Validation, Resources, Data Curation, Writing - Review & Editing

Stephan Henne - Software, Methodology, Software, Writing - Review & Editing

Ulas Im - Software, Validation, Resources, Data Curation, Writing - Review & Editing

Patricia K. Quinn - Methodology, Validation, Resources, Data Curation, Writing - Review & Editing, Funding acquisition
Lucia M. Upchurch - Methodology, Validation, Resources, Data Curation, Writing - Review & Editing, Funding
acquisition

Julia Schmale - Conceptualization, Methodology, Investigation, Resources, Writing - Review & Editing, Funding
acquisition, Supervision, Project administration

*Competing interests.* Eliza Harris is an Editor for Atmospheric Chemistry and Physics. The authors declare that they have no

other conflict of interest.




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
