# Peer review of "Data-driven modeling of environmental factors influencing Arctic methanesulfonic acid aerosol concentrations"

_EGUsphere, 2024_

## Author Comment (AC1)

The authors came up with data-driven models to simulate and predict the methanesulfonic acid in Arctic aerosol based on the observed data from 4 sites, and compared the models with traditional numerical models. I recommend publication after the following issues are addressed.

We thank the reviewer for their comments and suggestions. We have addressed each comment below with review comments in black, author response in blue, and additions to the original text in red. We have indented the author's response for clarity. Line numbers given in the author's response refer to lines in the revised manuscript.

1.  In "Abstract" Sec., line 24 and line 30, input features consider ambient conditions/ sources, chemical processing, and MSA removal. These two kinds of features do not include each other, it's better to make the expression unified.

To clarify, the input features for our data-driven models consider the ambient conditions experienced during air mass transport, rather than the in situ conditions measured at the measurement location. These include meteorological, oceanic, and geographical variables, listed in Table 2. In our interpretation of the output from these data-driven models, we identified three groups of features selected by the models to be important (sources, chemical processing, and removal). We have adjusted the text to clarify this.

Lines 24-25: In our approach, we create input features that consider the ambient conditions experienced during atmospheric transport (e.g., DMS emission, temperature, radiation, cloud cover, precipitation etc.)

Line 31-32: The data-driven models selected features which can be grouped into three categories, the sources, chemical processing, and removal of $MSA_p$.

2.  In "Introduction" Sec., lines 58-60, "which is enzymatically cleaved to produce DMS... in the atmosphere", better to write as "which is enzymatically cleaved to produce DMS, acting as the main source of marine atmosphere/aerosol".

We have amended the text according to the reviewer's suggestion.

Lines 61-62: Arctic marine phytoplankton and algae produce dimethylsulfoniopropionate as an osmoprotectant (Yoch, 2002), which is enzymatically cleaved to produce seawater DMS (Andreae, 1990; Kettle et al., 1999), which is the main source of marine biogenic sulfur in the atmosphere (Hulswar et al., 2022; Lana et al., 2011).

3.  In "Introduction" Sec., lines 68-69, the references should decrease to 2-3 classic or newest papers.

There is still uncertainty regarding the exact formation mechanisms of DMS oxidation into MSA, therefore, we list several modelling studies (Chen et al., 2018; Fung et al., 2022; von Glasow and Crutzen, 2004; Hoffmann et al., 2016) utilizing chemical transport and box models, a study from the Arctic utilizing observations and box modelling (Kecorius et al., 2023), an observational study from the Southern Ocean (Baccarini et al., 2021), and a study combing chamber measurements with box modelling (Wollesen de Jonge et al.,

2021). Therefore, we feel these references give the reader ample resources regarding DMS oxidation into MSA from several different perspectives.

4. In "Introduction" Sec., line 74, a repetition of lines 63-64, also line 277, please make it more concise.

The information on line 63-64 of the original manuscript describes the lifetime of gaseous DMS while the information on line 74 describes the lifetime of particulate MSA. These are two important pieces of information which was used in our decision of the duration of the air mass history on line 308-309 of the revised manuscript. We feel this information is vital to the reader and motivation for our methodological decision making, therefore we will keep this information as is.

5. In "Introduction" Sec., paragraph 2, the description of the relationship between DMA and MSA in the Arctic marine environment is presented in about 30 rows. Authors should adjust the proportion of this in the introduction, meanwhile, pay more attention to the description of numerical models and data-driven models applied in marine research and comparison of them, especially in the Arctic.

We have shortened the paragraph on DMS and MSA although Reviewer 2 wanted more detailed information about DMS oxidation processes to MSA, which we have added. We added more information about numerical modelling of DMS and MSA in the Arctic in the Introduction.

Lines 125-138: Many of these shortcomings are due to lack of knowledge concerning natural processes including rates and spatial distribution of DMS emission, oxidation mechanisms, and cloud processes. Ghahreman et al. (2017) showed that GEOS-Chem overestimated (underestimated) gaseous DMS in summer (spring) in the Canadian archipelago. The overestimation could be attributable to missing aqueous-phase oxidation mechanisms in GEOS-Chem while the underestimation in spring could be due to errors in the DMS source strength (Lana et al., 2011), with missing emissions from melt ponds and marginal ice zones (Gourdal et al., 2018; Hayashida et al., 2017; Mungall et al., 2016). Ghahreman et al. (2021) used the Global Environmental Multi-scale model–Modeling Air quality and Chemistry (GEM-MACH) model to demonstrate that the inclusion of DMS greatly improved the simulated size distribution compared to observations in the Arctic. However, errors in the parameterized nucleation mechanisms led to discrepancies for particles smaller than 50 nm, having implications for cloud formation as aerosols of these sizes have been shown to activate in the Arctic (Leaitch et al., 2016). Hoffmann et al. (2021) was able to improve simulations of gaseous MSA in the ECHAM-HAMMOZ model by implementing aqueous-phase oxidation mechanisms on deliquesced particles and by considering the reactive uptake of methanesulfinic acid (MSIA). However, in-cloud processing of MSA is still missing from this model configuration and reactive uptake coefficients are not well parameterized as they depend on aerosol acidity, thus further improvements are required.

6. In "Results" Sec. 3.2, lines 610-622, this paragraph should move to Methods.

We have moved lines 610-617 of the original manuscript to the Sect. 2.7 of the Methods (lines 560-567 of the revised manuscript), we have kept the text on lines 617-622 of the original manuscript in place to state our intentions for this comparison and introduce the section.

7.    In "Results" Sec. 3.3, line 743, how to convince readers of the "high accuracy" of max R2=0.29, may be compared to numerical models, it is better to change an appropriate expression of it. Fig.9 is good evidence, maybe move the expression to Fig. 9 discussing.

To clarify, the intention of this text is not to state that our data-driven models have "high accuracy" but that they "struggle to reproduce observed MSAp with high accuracy". We can see how this formulation of the text can create confusion. We have amended the text to clarify this.

Lines 807-808: While our data-driven models struggle to accurately reproduce the observed $MSA_p$ (max $R^2$ = 0.29), they can capture the variability (PCC up to 0.77), and they outperform the classic numerical models.

8.   In "Results" Sec. 3.3, line 746, there is no performance comparison between numerical and data-driven models in Fig. 4, only data-driven models including RF and AM, please confirm.

We thank the reviewer for this good catch. We have added text to indicate the numerical models are compared against in situ observations in Figs. 3, S2, S3, and S4 and the evaluation metrics for the data-driven models are given in Fig. 4.

Lines 809-811: This is evident from a comparison of the negative $R^2$ values for the numerical models (Figs. 3, S2, S3, and S4), indicating the numerical models are worse at predicting $MSA_p$ compared to the mean of the observations, versus the evaluation metrics for the data-driven models (Fig. 4).

9.    In "Results" Sec. 3.3, line 750, it is confusing to mention Fig. 9 and Fig. 10 here, not to quote the latter figures when discussing the present figure.

We have moved this text to lines 1207-1209 of the revised manuscript to avoid discussing the figures out of order.

Andreae, M. O.: Ocean-atmosphere interactions in the global biogeochemical sulfur cycle, Mar. Chem., 30, 1–29, https://doi.org/10.1016/0304-4203(90)90059-L, 1990.

Ghahreman, R., Norman, A.-L., Croft, B., Martin, R. V., Pierce, J. R., Burkart, J., Rempillo, O., Bozem, H., Kunkel, D., Thomas, J. L., Aliabadi, A. A., Wentworth, G. R., Levasseur, M., Staebler, R. M., Sharma, S., and Leaitch, W. R.: Boundary layer and free-tropospheric dimethyl sulfide\hack\break in the Arctic spring and summer, Atmospheric Chem. Phys., 17, 8757–8770, https://doi.org/10.5194/acp-17-8757-2017, 2017.

Ghahreman, R., Gong, W., Beagley, S. R., Akingunola, A., Makar, P. A., and Leaitch, W. R.: Modeling Aerosol Effects on Liquid Clouds in the Summertime Arctic, J. Geophys. Res. Atmospheres, 126, e2021JD034962, https://doi.org/10.1029/2021JD034962, 2021.

Gourdal, M., Lizotte, M., Massé, G., Gosselin, M., Poulin, M., Scarratt, M., Charette, J., and Levasseur, M.: Dimethyl sulfide dynamics in first-year sea ice melt ponds in the Canadian Arctic Archipelago, Biogeosciences, 15, 3169–3188, https://doi.org/10.5194/bg-15-3169-2018, 2018.

Hayashida, H., Steiner, N., Monahan, A., Galindo, V., Lizotte, M., and Levasseur, M.: Implications of sea-ice biogeochemistry for oceanic production and emissions of dimethyl sulfide in the Arctic, Biogeosciences, 14, 3129–3155, https://doi.org/10.5194/bg-14-3129-2017, 2017.

Hoffmann, E. H., Heinold, B., Kubin, A., Tegen, I., and Herrmann, H.: The Importance of the Representation of DMS Oxidation in Global Chemistry-Climate Simulations, Geophys. Res. Lett., 48, e2021GL094068, https://doi.org/10.1029/2021GL094068, 2021.

Hulswar, S., Simó, R., Galí, M., Bell, T. G., Lana, A., Inamdar, S., Halloran, P. R., Manville, G., and Mahajan, A. S.: Third revision of the global surface seawater dimethyl sulfide climatology (DMS-Rev3), Earth Syst. Sci. Data, 14, 2963–2987, https://doi.org/10.5194/essd-14-2963-2022, 2022.

Kettle, A. J., Andreae, M. O., Amouroux, D., Andreae, T. W., Bates, T. S., Berresheim, H., Bingemer, H., Boniforti, R., Curran, M. A. J., DiTullio, G. R., Helas, G., Jones, G. B., Keller, M. D., Kiene, R. P., Leek, C., Levasseur, M., Malin, G., Maspero, M., Matrai, P., McTaggart, A. R., Mihalopoulos, N., Nguyen, B. C., Novo, A., Putaud, J. P., Rapsomanikis, S., Roberts, G., Schebeske, G., Sharma, S., Simó, R., Staubes, R., Turner, S., and Uher, G.: A global database of sea surface dimethylsulfide (DMS) measurements and a procedure to predict sea surface DMS as a function of latitude, longitude, and month, Glob. Biogeochem. Cycles, 13, 399–444, https://doi.org/10.1029/1999GB900004, 1999.

Lana, A., Bell, T. G., Simó, R., Vallina, S. M., Ballabrera-Poy, J., Kettle, A. J., Dachs, J., Bopp, L., Saltzman, E. S., Stefels, J., Johnson, J. E., and Liss, P. S.: An updated climatology of surface dimethlysulfide concentrations and emission fluxes in the global ocean, Glob. Biogeochem. Cycles, 25, GB1004, https://doi.org/10.1029/2010GB003850, 2011.

Leaitch, W. R., Korolev, A., Aliabadi, A. A., Burkart, J., Willis, M. D., Abbatt, J. P. D., Bozem, H., Hoor, P., Köllner, F., Schneider, J., Herber, A., Konrad, C., and Brauner, R.: Effects of 20–100 nm particles on liquid clouds in the clean summertime Arctic, Atmospheric Chem. Phys., 16, 11107–11124, https://doi.org/10.5194/acp-16-11107-2016, 2016.

Mungall, E. L., Croft, B., Lizotte, M., Thomas, J. L., Murphy, J. G., Levasseur, M., Martin, R. V., Wentzell, J. J. B., Liggio, J., and Abbatt, J. P. D.: Dimethyl sulfide in the summertime Arctic

atmosphere: measurements and source sensitivity simulations, Atmospheric Chem. Phys., 16, 6665–6680, https://doi.org/10.5194/acp-16-6665-2016, 2016.

Yoch, D. C.: Dimethylsulfoniopropionate: Its Sources, Role in the Marine Food Web, and Biological Degradation to Dimethylsulfide, Appl. Environ. Microbiol., 68, 5804–5815, https://doi.org/10.1128/AEM.68.12.5804-5815.2002, 2002.

---

## Author Comment (AC2)

Pernov et al. presents a data-driven approach to model particulate methanesulfonic acid (MSAp) concentrations in the Arctic using Random Forest (RF) and Additive Models (AM). The work is interesting because it combines several data sources and uses careful feature engineering to capture different environmental condition. I would suggest the paper to be published after improvement of the following points by the authors to make their discussion deeper:

> We thank the reviewer for their comments and suggestions. We have addressed each comment below with review comments in black, author response in blue, and additions to the original text in red. We have indented the author's response for clarity. Line numbers given in the author's response refer to lines in the revised manuscript.

1. The paper mentions the atmospheric oxidation of dimethyl sulfide (DMS) to form MSAp but does not go into detail about how gas-phase and aqueous-phase oxidation work. I would suggest the authors to add a brief discussion on these processes and explain how they change under different Arctic conditions.

> The reviewer is correct that we could have added more details about the mechanisms of gas and aqueous phase oxidation of DMS into $MSA_p$. Therefore, we have added the following lines about the mechanisms, intermediates, and unique aspects relevant to the Arctic.
>
> Lines 66-87: DMS oxidation to $MSA_p$ involves a variety of gas- and aqueous-phase reactions, the latter occurring in cloud droplets or on deliquesced particles (Barnes et al., 2006a; Chen et al., 2018; Fung et al., 2022; Hoffmann et al., 2016). DMS is first oxidized through two major branches. One is the abstraction pathway by reactions with OH, $NO_3$, and Cl radicals in the gas-phase to yield methylthiomethylperoxy radical (MTMP: $CH_3SCH_2OO$) (Berndt et al., 2019; Hoffmann et al., 2016). MTMP can undergo isomerization to form hydroperoxymethylthioformate (HPMTF) (Berndt et al., 2019; Veres et al., 2020), or oxidation by NO or $RO_2$ to produce $CH_3SO_2$, which can then form $SO_2$, sulfuric acid ($H_2SO_4$), or MSA with strongly temperature-dependent yields (Berndt et al., 2023; Chen et al., 2023; Shen et al., 2022). The other DMS oxidation branch is the addition pathway through reactions with OH, BrO, Cl, as well as $O_3$ to yield dimethyl sulfoxide (DMSO), which occurs mainly through gas-phase but partly in the aqueous-phase through reaction with $O_3$ (Hoffmann et al., 2016). DMSO is a semi-volatile species which reacts with OH in both gas- and aqueous-phase to form methanesulfinic acid (MSIA). MSIA then reacts with OH or $O_3$ in the aqueous-phase to produce $MSA_p$ (Chen et al., 2018; von Glasow and Crutzen, 2004; Hoffmann et al., 2016; Wollesen de Jonge et al., 2021), although it can also undergo gas-phase oxidation by OH to yield $CH_3SO_2$ and thus contributing to the temperature-dependent pathway to produce gaseous MSA (Chen et al., 2023;

Shen et al., 2022). The produced gas-phase MSA can condense on cloud droplets or existing particles to form $MSA_p$ (Hoffmann et al., 2016). Aqueous phase reactions are dominant formation mechanisms for $MSA_p$ in a typical marine boundary layer condition (Baccarini et al., 2021; Chen et al., 2018; Hoffmann et al., 2016; Kecorius et al., 2023). In the Arctic, cold temperatures (Barnes et al., 2006a; Chen et al., 2023; Shen et al., 2022) and elevated halogen levels will favor $MSA/MSA_p$ formation relative to $SO_2$ (Chalif et al., 2024; Jongebloed et al., 2023; Sørensen et al., 1996) especially in the springtime, through both gas- and aqueous-phase pathways. The presence of ice containing clouds may limit the aqueous-phase production since reactions between DMS and its intermediates and oxidants are mainly able to occur at the surface (Chen et al., 2018). For a full description of DMS oxidation mechanisms and pathways see Barnes et al.(2006a).

2. The paper uses RF and AM to analyze the data, but I would suggest the authors to discuss more clearly the strengths of these methods. For example, RF is useful for capturing non-linear relationships and provides feature importance, while AM offers a simple way to understand how each variable affects MSAp. A clear discussion on these points would help readers see why these methods were chosen.

We have added a new Section (2.6.3) to clarify and discuss further the respective strengths and limitations of both RF and AM approaches. Also, Supplementary Text 2 and Figure S1 provide summarized results and explanations about other approaches we tested. As explained in the first paragraph of Section 2.6, these were not retained mainly because they performed worse in terms of prediction error.

Lines 540-558 (new Section 2.6.3 titled Strengths and Limitations of RF and AM): Both RF and AM are non-linear regression models although they differ in a few key aspects. First, RF's output is the average of predictions from many decision trees which are based on random subsets of the training data and random subsets of features, while AM considers the entire training set at once (i.e., without random subsets). This randomization generally reduces the risk of overfitting (yielding a smaller prediction variance) compared to constructing a single, large decision tree. For AM, the risk of overfitting was minimized by keeping the number of splines bases low ($J$=5) and enforcing this stepwise variable selection scheme so that the number of parameters $P$ stayed relatively low. In that sense, AM is generally a simpler model than RF. Second, the predictions from decision trees, and thus from RF, seen as a mathematical mapping from a feature space to a target variable space, are piecewise constant functions. By contrast, the predictions from AM are by design smooth, as they are computed as the sum of cubic splines (with

continuous second derivatives). In practice, this means that the predicted target surface from RF looks like jagged stairs, with jumps at feature splits, while for AM this looks like a smooth surface. Finally, the additive structure of AM in Eq. (4) is quite constrained: features have their respective effects, and these add up to a prediction. We considered pairwise interactions but no higher-level terms (e.g., three-way interactions). By comparison, RF inherently can include higher level interactions, as splits are being added sequentially (i.e., conditioned on previous splits) when growing a decision tree (up to a maximum depth, which we tuned as a hyperparameter). This higher complexity makes RF generally more flexible than AM. Although this flexibility comes at the cost of harder interpretability as one cannot easily visualize the estimated effect of a feature on the target, specifically because of such interactions likely being different from tree to tree within the ensemble. The additive constraint of AM is what makes the estimated partial effects directly interpretable, say, as a curve displayed in a plot.

3. While the authors note the low $R^2$ values, I would suggest they discuss this further. They could explain which important variables might be missing and how these missing elements or measurement issues might affect the results. I would also suggest discussing future improvements, like using more advanced methods or combining physical models with data-driven approaches.

We have added some discussion about possible reasons for the low evaluation metrics. A common theme for all models and feature groups tested is that they under/over-predict high/low observations, therefore, we have added an example of how the models could under/over-predict high/low observations based on inadequacies of the input features.

Lines 762-767: For instance, models underpredict high observations of $MSA_p$ which could be due to large DMS emissions not being captured in the input features either due to being based on a climatology of seawater DMS (Lana et al., 2011; Nightingale et al., 2000) or occurring further back in time than 5 days. The models also overpredict low $MSA_p$ observations, which could be due to extreme precipitation events not being captured by ERA5 (Loeb et al., 2022) or being smoothed out in the feature engineering procedure. Although the summation was used as an aggregation method for precipitation therefore smoothing over one day should not affect extreme events (Table 2).

We have also added some text adding to our discussion of potential improvements to data-driven modeling for Arctic $MSA_p$ predictions in the Conclusions.

Lines 1234-1244: When sufficient long-term high-resolution data becomes available, leveraging the power of other data-driven approaches (e.g., neural networks) could be an option for advancing data-driven modeling of Arctic aerosols. However, when limited by the sample size, less complex tree-based models and statistical methods can often perform on par or better than neural networks (Grinsztajn et al., 2022). One option would be to combine data driven modeling with physically constrained loss mechanisms, known as physics-informed neural networks (PINN) (Cuomo et al., 2022). This avenue could help ensure the proper ingredients (precursors, oxidants, meteorology, etc.) are present at sufficient levels and with the correct temporal occurrence. The multi-input/output functionality of neural networks could also help elucidate the branching ratio of DMS oxidation mechanisms and the partitioning between gaseous and particulate phase MSA. However, satisfactory long-term high resolution, concurrent measurements of gas-phase DMS, gas-phase MSA and particulate phase MSA need to become available at appropriate locations dispersed around the Arctic region, which remains a challenge both logistically and monetarily.

We have added text to show that aerosol chemical and physical properties are missing from our input features and how this would affect MSA partitioning between gas and condensed phase in the Conclusion.

Lines 1251-1256: Another missing component from our feature list is aerosol chemical and physical properties (e.g., surface area, mixing state, hygroscopicity, and composition), which largely determines the reactive uptake of gaseous MSA (and its intermediates) onto preexisting aerosols (Dada et al., 2022; Yan et al., 2020). Acidic and effloresced aerosols are less likely to uptake gaseous MSA while alkaline and deliquesced aerosols are more likely, including these parameters in future data-driven approaches could help resolve the equilibrium partitioning between gas and condensed phase MSA, thus representing another sink term for $MSA_p$.

4. The paper explains how the models work but could do more to show how each feature relates to the physical processes in the Arctic. I would suggest the authors to compare the RF feature importance with the partial effects from the AM, so readers can see the real-world significance of the results.

In Sect 3.4, we detail the features selected by each model and for each group of variables. We also describe how each model selects features. We show that each model selects themes of variables (source related, chemical processing, and removal) for most stations. We also discuss the differences between the variable

groups (Group A and A+B) selected by the models. In the following sections, we extensively discuss the source related, chemical processing, and removal themes of features selected as well as other features such as physical meteorology related and vertically oriented features. In the main text, we describe the partial effects and their physical interpretations for Alert and Utqiaġvik/Barrow and summarize the remaining stations. In the Supplemental Information, we go into detail on the partial effects for the other station/dataset combinations. Therefore, we feel we have adequately described how the features selected by the models relate to physical processes in the Arctic. The reviewer suggests comparing the RF feature importance, which is a single number or global importance value, to the relationship displayed by the partial effects plots (or local importance values) from the AM. We explored partial dependency plots (which is a local feature importance method) for RF and ultimately opted not to present the results since they were similar to the partial effects from the AM although less reliable. While the AM partial effects work in a similar manner to partial dependency plots from a RF model, by keeping all other features at their average while varying a single feature across its distribution range to examine the relationship with model output, we are more confident in the AM results due to differences in features selection and model design (increased interpretability compared to RF, see Sect. 2.6.3). RF struggles with explainability when features are strongly correlated due to the random feature sub-selection during tree construction, this makes consistently determining feature importance difficult. AM does not struggle with correlated features due to selecting them sequentially and by considering the entire dataset of features once selected. This is evident by RF selecting features of the same variable type but with different timestamps compared to AM (Table 4). This would make the interpretation of partial dependency plots more complex compared to the partial effects of AM, whose feature selection and partial effects are more straightforward and reliable. This is due to the additive constraint of AM, which makes the partial effects directly interpretable. The AM was tailored for this specific project while the RF served as a baseline model for comparison. Therefore, we opted to only present results for the partial effects of AM in the manuscript.

[revised manuscript text omitted]